# The spatial dimension in the assessment of urban socio-economic vulnerability related to geohazards, a systematic review

5  Diana Contreras[1,2], Alondra Chamorro[1], Sean Wilkinson[2]

[1]Research Center for Integrated Disaster Risk Management (CIGIDEN), Pontifical Catholic University, Santiago, 7820436, Chile

[2] School of Engineering, Newcastle University, Newcastle upon Tyne, NE1 7RU, United Kingdom (UK)

*Correspondence to*: Diana Contreras (diana.contreras@cigiden.com – diana.contreras-mojica@newcastle.ac.uk)

**Abstract**

Society and economy are only two of the dimensions of vulnerability. This paper aims to elucidate the state of the art in data sources, spatial variables, indicators, methods, indexes and tools for the spatial assessment of socio-economic vulnerability

15  (SEV) related to geohazards. This review was first conducted in December 2018 and re-run in March 2020 for the period between 2010 and 2020. The gross number of articles reviewed were 27, from which we identified 18 relevant references using a revised search query, and six relevant references identified using the initial query giving a total sample of 24 references. The most common source of data remains population census. The most recurrent spatial variable used for the assessment of SEV is households without basic services, while critical facilities are the most frequent spatial categories. Traditional methods have

20  been combined with more innovative and complex methods to select and weight spatial indicators and develop indices. The Social Vulnerability Index (SoVI®) remains the benchmark for the assessment of SEV and a reference for its spatial assessment. Geographic information systems (GIS) is the most common tool for conducting a spatial assessment of SEV regarding geohazards. For future spatial assessments of SEV regarding geohazards, we recommend considering 3D spatial indexes at the microscale in the urban level involving the community in the assessments.

## 1 Introduction

Vulnerability is defined by United Nations (UN) as 'the conditions determined by physical, social, economic and environmental factors or processes which increase the susceptibility of an individual, a community, assets or systems to the

impact of hazards' (UN, 2016). In the past, vulnerability was considered a composite factor having only two dimensions: exposure to risk and susceptibility (Béné, 2009; Chambers, 1989). More recently, Birkmann (2013) considered three factors: exposure, susceptibility, and fragility and lack of resilience. The degree of vulnerability of a specific community is a human value judgement that highly influences management decisions (McLaughlin et al., 2002). In addition, the concept of social

vulnerability (SV) to environmental hazards involves demographic and socio-economic factors that affect community resilience (Zebardast, 2013), and this is considered a hot topic in current disaster research (Shen et al., 2018). The social and economic dimensions are only two dimensions of vulnerability to multiple stressors and shocks. These shocks include disasters due to the fragility and susceptibility of human well-being damaged by disruption to individuals (physical and mental health) and collective social systems (e.g., education, services, health) and their characteristics (e.g., age, ethnicity, disabilities)

(Birkmann et al., 2013). Social vulnerability refers to the inability of people, organisations, and societies to cope with negative impacts from different stressors to which they are exposed (Eidsvig et al., 2014; Kuhlicke et al., 2011; Myers et al., 2008; Qasim et al., 2018). Typically, this inability results from pre-existing conditions that reduce a society's ability to prepare and recover from disasters (Alcorn et al., 2013; Cutter and Finch, 2008; Eidsvig et al., 2014; Zebardast, 2013; Zhou et al., 2014). Social vulnerability additionally identifies sensitive populations that are less prepared to respond, cope with, and recover from

a disaster (Zebardast, 2013) such as low-income populations, women, pregnant women, children below 5 years, elderly above 65 years (Bereitschaft, 2017a; Zhou et al., 2014) and physically and or mentally challenged individuals (Contreras and Kienberger, 2012). Other vulnerable population groups are people with language, cultural and spatial barriers (Eidsvig et al., 2014) such as migrants (Yuan et al., 2019a), rural population, people without post-secondary education (Bereitschaft, 2017a; Cutter et al., 2003; Eidsvig et al., 2014), high-density population (Cutter et al., 2003; Eidsvig et al., 2014) and public transport

captives (Bereitschaft, 2017a).

The concept of SV is complex and dynamic, changing through the time and over space, and therefore not easily captured by a single variable (Cutter and Finch, 2008; Zebardast, 2013). It represents the multidimensionality of disasters by focusing attention on the totality of relationships in a given social situation, which, in combination with environmental forces, such as

geohazards, result in a disaster (Oliver-Smith, 2003). Social vulnerability attracts less attention by researchers because many

challenges are implied in its quantification (Qasim et al., 2018). Power relationships that exclude certain individuals or groups from benefiting from disaster risk reduction (DRR) or post-disaster recovery efforts are examples of SV (Contreras et al., 2011). These power relationships manifest between individuals or socio-economic groups in the framework of institutions or culturally determined dialogues about stressors (Warmer et al., 2007).

The economic dimension of vulnerability is the predisposition for the loss of economic value from damage to physical assets (Birkmann et al., 2013) and/or business interruption (activities, services or delivery of products). The assessment of SV is orientated to cast light on the most susceptible groups of a population to be impacted by a disaster, in the spatial and temporal dimensions (Zhou et al., 2014). Another important aspect to consider is the relationship between social and economic

dimensions because, according to Noy (2009), no evidence exists of a correlation between consequences of disasters, such as the number of fatalities or affected population, and GDP growth. Nevertheless, the same author indicates that the degree of damage due to a disaster will negatively influence GDP growth. Thus, Noy (2015) proposes to integrate the number of fatalities and injuries with financial damage due to a disaster using a model similar to the estimation of disability-adjusted life years (DALYs). His index accounts for the number of human years lost as a result of the damage. The spatial dimension of socio-

economic vulnerability (SEV) recognises that people and groups of similar characteristics tend to occupy the same or similar areas, while the temporal dimension of SEV makes reference to people's degree of vulnerability that can change depending on age, life situation, and season (Wisner and Uitto, 2009). To include urban vulnerability assessment into a spatial plan requires strategic, technical, substantial, and procedural integration (Hizbaron et al., 2012). According to Ebert et al., (2009) a spatial indicator of SV is an SV indicator with a physical component. Housing structures and the built environment were

previously included by Shuang-Ye, Brent, and Ann (2002) in a GIS-based study of SV. The link between transportation infrastructure and land use had been already studied by Clark et al. (1998). The physical conditions were considered indicative of the social ones by Rashed and Weeks (2003). Kienberger et al.,(2009) proposed a methodology for the spatial quantification of vulnerability and the identification of vulnerability units build upon the *geon* concept, which is a framework for the clustering of homogeneous spatial information. Khazai et al., (2013) developed a sector-specific vulnerability index (IVIs),

which included transport dependency indicators made up by the spatial variables such as freight transport volume road and

freight transport volume railway; this index also included the spatial variable of customer proximity as part of the indicator demand dependency.

In the context of disaster risk management, and mainly for exposure and impact assessment, the accuracy and reliability of input data are two of the most important factors (Aubrecht et al., 2013). Data constraints play a key role in the results of the SEV assessment, with the number of variables changing the assessment and the inclusion of additional variables enhancing its precision and enabling the proper presentation of SV assessment (Gautam, 2017). Thus, the assessment of vulnerability must be based on indicators and proxy indexes (Qasim et al., 2018) that can guarantee objectivity and can provide quantitative metrics to compare different places (Cerchiello et al., 2018). Indicators and indexes are defined as single qualitative or indirect quantitative measures of a characteristic (Chen, 2016) or a real phenomenon (Fekete, 2009) resulting from systematically observed facts (OECD, 2008). Indicators transform complex data into manageable units of information for performance, change, and achievement assessment (Grace and Edwin, 2009). Indicators also summarise technical information into indexes, simplifying comprehension (Simpson and Katirai, 2006). The most important factor for indicator selection is the availability of data. The lack of data can lead to reliance on variables that may not be the most accurate indicators of vulnerability (Zhou et al., 2014). Vulnerability indicators are complex measures of a part of what constitutes a community. Scientific literature has identified groups of social and economic indicators, which combined with physical and land data, are useful for the vulnerability assessment of communities (King, 2001). The use of these indicators has primarily been applied to the assessment of adaptive capacity and vulnerability (Chen, 2016).

Indexes are built up with those indicators and later mapped to display the different categories of vulnerability in each administrative zone, limiting the spatial dimension to this stage. The construction of an index implies selection of indicators, indicator normalization and weighting, and aggregation into an index (OECD, 2008) that must collectively represent aspects of a society's ability to prepare for, deal with, and recover from a disaster (Eidsvig et al., 2014). The most sensitive step for constructing an index is the weighting of indicators. This can be undertaken either using participatory approaches such as the analytic hierarchy process (AHP), the budget allocation process, statistical assessment like the principal component analysis (PCA), or factor analysis (FA) (Eidsvig et al., 2014; OECD, 2008). Weighting individual indicators is a major challenge for

constructing a composite indicator for vulnerability (Adger et al., 2004; Zebardast, 2013). The objectives of indicators weighting are first, to investigate any correlation among indicators to detect overlapping information and second, to select a suitable weighting and aggregation approach for the final index calculation. Different weightings show varied spatial vulnerability patterns (Papathoma-Kohle et al., 2019); however, independent of the method applied, after comparing 106 studies for index construction with respect to risk assessment, Beccari (2016) found that the most common approach used (41.5%) was the 'equal weights' method. Eventually, the accuracy of SV assessment lies on the accuracy of input data (Yuan et al., 2019a) and not on the weighting method. After being weighted, indicators can be aggregated using additive, multiplicative, or decision rule models (Eidsvig et al., 2014). The method of aggregation is one of the most pressing problems in developing composite vulnerability indices (Rygel et al., 2006).

Composite indicators have been commonly employed by researchers, planners, and disaster managers for vulnerability assessments (Yuan et al., 2019a). Cutter, Boruff and Shirley (2003) have constructed an index of SV called (SoVI®) for environmental hazards in the United States using a factor analytic approach computed in a summary score based on an additive model. In the framework of the Methods for the Improvement of Vulnerability Assessment in Europe (MOVE) project, variables were grouped into single (Vinchon et al., 2011) and composite indicators. In the case study area of Salzburg (Austria), an expert-based approach was chosen, and several experts were asked to allocate weights according to the contribution of each variable to the vulnerability of floods (Contreras and Kienberger, 2011). Other composite indicators useful for the vulnerability assessment are the Prevalent Vulnerability Index (Cardona, 2005), Environmental Sustainability Index (Esty et al., 2005), and Human Development Index (UNDP, 2010). All these indexes face challenges when assessing vulnerability indicators, such as: ranking socio-economic data on an interval scale, dealing with temporal aspects (day-night changes), choosing the most suitable data resolution to avoid the 'modifiable areas unit problem' (MAUP) (Openshaw, 1983), deciding how to allocate a meaningful value to socio-economic variables, and how these aspects together affect the vulnerability assessment of each case study areas (McLaughlin et al., 2002). The compilation of all of the SV indicators used through time was undertaken by Fatemi, Ardalan, Aguirre, Mansouri and Mohammadfam (2017); however, they neither included the spatial dimension in their systematic review nor focused exclusively on geohazard as in this research.

Quantitative measures to develop indicators can be spatially explicit and based on spatial variables, such as location, area, range, distance, direction, spatial geometries, and patterns (Unwin, 1996), spatial connectivity, mobility (Béné, 2009), isolation, diffusion, distribution, spatial association, spatial interaction, spatial evolution, spatial synthesis and scale of the affected area, and surroundings (Béné, 2009; Buzai and Villerías Alarcón, 2018; Contreras et al., 2013; Meentemeyer, 1989).

The geographic patterns in vulnerability can increase due to spatial interactions; while additional patterns within these components may be related to the nature of vulnerability stemming from a specific hazard (Amram et al., 2011). The main aim of this research is to elucidate the state of the art in data sources, spatial variables, indicators, methods, indexes, and tools for the assessment of the SEV related to geohazards in urban environments. Geohazards can be endogenic such as earthquakes, tsunamis, and volcanic eruptions and exogenic such as landslides, soil erosion, and land degradation. We particularly focus on

these phenomena for two reasons: first, geohazards are the natural phenomena that have produced the highest quantity of losses in recent years in the urban environments, particularly earthquakes, and, second, because geohazards are the phenomena addressed by the institutions involved in the present research.

The Indian Ocean tsunami in 2004, as a result of its large impact area, reignited the research community's interest in spatial

vulnerability analyses, illuminating the problems faced by low-income population after disasters (Fekete, 2012). This approach was aligned with the Hyogo Framework for Action (UNISDR, 2007), and confirmed by Gautam (2017), who notes that after 2005 a focus on construction and mapping of the SV index intensified. Thus, the use of geographic information systems (GIS) to collect and process data related to hazards and vulnerability was found very suitable (Fekete, 2012). Major earthquakes that occurred during the same period as this systematic review (2010-2020), e.g., Chile (2010), New Zealand (2010 and 2011),

Nepal (2015), Mexico (2017), Albania (2019), and Croatia (2020) demonstrate the vulnerability of urban areas to seismic damages (Armaş et al., 2017).

This research reviews case study areas, data sources, spatial variables, indicators, methods, indexes, and tools used in the spatial assessment of SEV vulnerability by different authors in the period between 2010 and 2020. This systematic review

aims to evaluate the literature to identify patterns and trends, as well as research gaps, to recommend new research areas. This

article aspires to guide scientists who want to perform any spatial assessment of SEV vulnerability. Socio-economic vulnerability is dynamic and changes across spatial and temporal scales, depending on demographic, geographic, economic, and cultural factors. Hence, no one-size-fits-all approach exists to measure and reduce SV (Zhou et al., 2014). This paper is divided into six sections. The introduction is the first section and includes a literature review. The second section, on methods,

elaborates on the criteria for selecting the articles that comprise the systematic review and the format of the presentation of results. The third section focuses on the results. The fourth section includes discussion of the results supported by literature, and, the fifth section contains conclusions with recommendations proposed in the sixth section.

## 2 Methods

A systematic review searches for, appraises, and synthesises research evidence (Grant and Booth, 2009). In the present

research, the systematic review was conducted to elucidate the state of the art of data sources, spatial variables, indicators, methods, indexes and tools for the spatial assessment of the SEV related to geohazards, which we consider is covered in the period between 2010 and 2020. Thus, the main research question is: what is the state of the art in the spatial assessment of SEV to geohazards in urban environments?

This review was conducted in December 2018 and re-run during the revision process in March 2020. For this research, Clarivate Analytics and Scopus/Elsevier were the sources of selected literature given their functionalities to run the search query. We limited the query to articles published in academic journals because they typically are rigorous in the selection of their publications and therefore contain a complete and accurate description of methodologies and consistent results. The terms selected for the search query refer to vulnerability in the socio-economic dimension, the spatial variables listed by

Meentemeyer (1989), Béné (2009), Contreras et al. (2013) and Buzai and Villerías Alarcón (2018) and the aforementioned endogenic and exogenic geohazards. Based on several screenings, to refine the search strategy, we opted to exclude terms that were not related to geohazards and were recurring in the titles, abstracts and keywords of the resulting references. The final set of terms included and excluded in the search query are listed in Table 1 and the scheme of the methodology applied is depicted in Figure 1.

| D | Q | SEARCH TERMS |
|---|---|---|
| Clarivate analytics | TOPIC | "social vulnerability" OR "economic vulnerability" OR "socioeconomic vulnerability" OR "socio-economic vulnerability" |
| | AND | |
| | TOPIC | "area" OR "distance" OR "range" OR "distance" OR "direction" OR "spatial geometries" OR "patterns" OR "spatial connectivity" OR "isolation" OR "diffusion" OR "spatial association" OR "scale" OR "accessibility" OR "network" OR "cluster" |
| | AND | |
| | TOPIC | "earthquakes" OR "tsunamis" OR "volcanic eruptions" OR "landslides" OR "soil erosion" OR "land degradation" |
| | NOT | |
| | TOPIC | "climate change" OR "ecological" OR "drought" OR "resilience" OR "debris" OR "epidemiological" OR "substance" OR "behavioural" OR "evacuation" OR "recovery" OR "pollution" OR "leptospirosis" OR "violence" OR "illness" OR "disease" OR "heat" OR "crisis" OR "conflict" OR "deaths" OR "obesity" OR "criminal" OR "chemical" OR "symptoms" OR "syndrome" OR "food insecurity" OR "air pollution" OR "stress" OR "diabetes" OR "depressive" OR "alcohol" OR "cancer" OR "drugs" OR "palm oil" OR "tobacco" OR "smoke" OR "storm" OR "psychometric" OR "cocaine" OR "toxic" OR "palliative" OR "therapy" OR "HIV" OR "dengue" OR "ecosystem" OR "rheumatoid" "arthritis" OR "nutritional" OR "malaria" OR "resources" OR "sexual activity" OR "sexual health" |
| Scopus/Elsevier | Article title, abstract, keywords | (TITLE-ABS-KEY ( "social vulnerability*" AND "economic vulnerability*") AND TITLE-ABS-KEY ("socioeconomic vulnerability*") AND TITLE-ABS-KEY ("area" OR "distance" OR "range" OR "distance" OR "direction" OR "spatial geometries" OR "patterns" OR "spatial connectivity" OR "isolation" OR "diffusion" OR "spatial association" OR "scale" OR "accessibility" OR "network" OR "cluster") AND TITLE-ABS-KEY ("earthquakes" OR "tsunamis" OR "volcanic eruptions" OR "landslides" OR "soil erosion" OR "land degradation") AND NOT TITLE-ABS-KEY ("climate change" OR "ecological" OR "drought" OR "resilience" OR "debris" OR "epidemiological" OR "substance" OR "behavioral" OR "evacuation" OR "recovery" OR "pollution" OR "leptospirosis" OR "violence" OR "illness" OR "disease")) AND DOCTYPE (ar) AND PUBYEAR > 2009 AND PUBYEAR <2021 |

D: Database

Q: Query

Table 1. Terms included and excluded to identify relevant literature references.

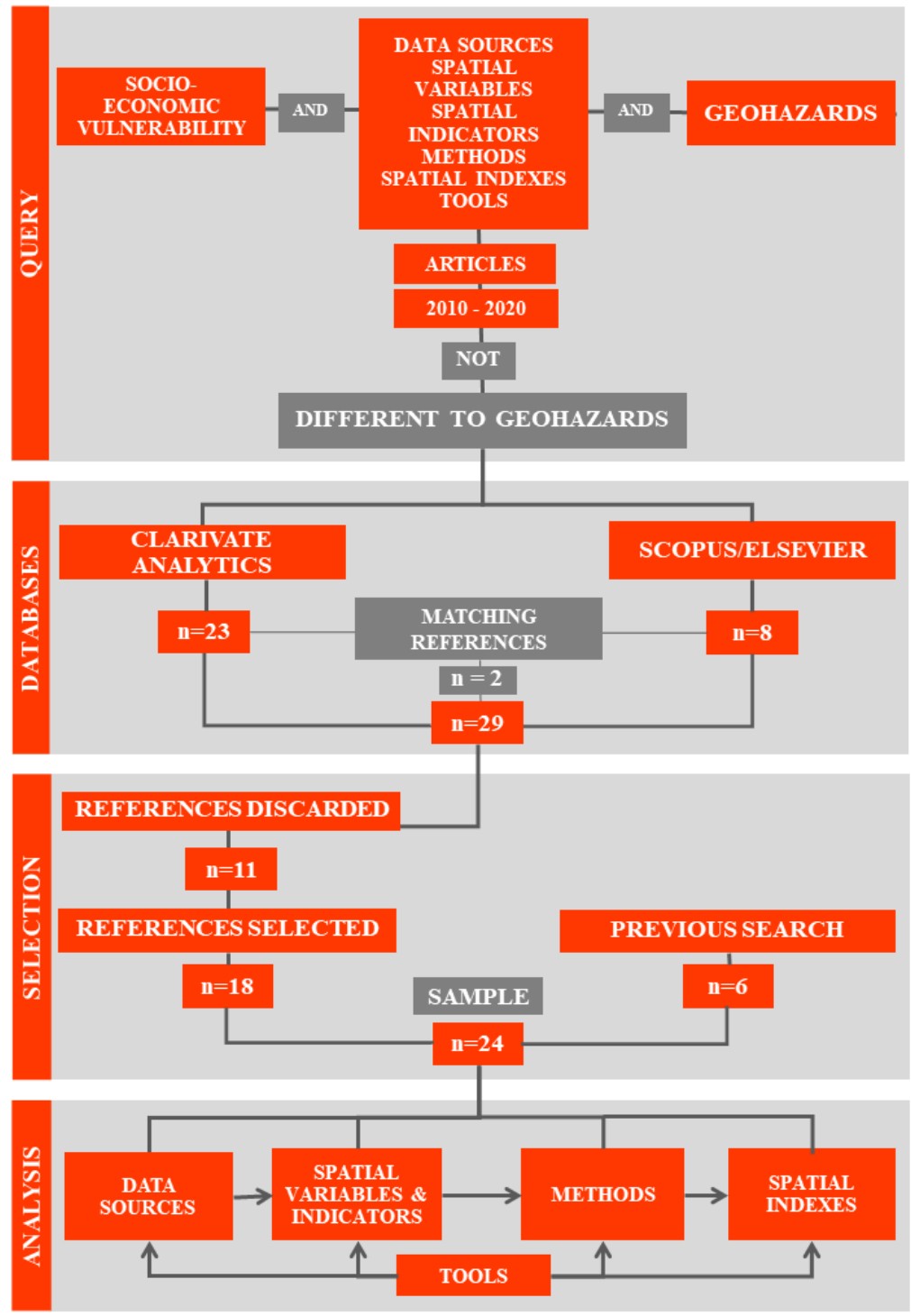

Figure 1. Methodology applied for the systematic literature review.

The findings will be presented in the results section in tables related to selected references, data sources, spatial variables, indicators, methods, spatial indexes, and tools. Table 2 is structured in four columns, namely author, year, research objective, geohazard addressed, and country where the case study area of the paper is located. The authors are listed from the most recent reference to the oldest one. Tables 3, 4, 5, 6 and 7 are structured mainly in two columns: the first column lists data sources, spatial variables, indicators, methods and indexes respectively. The second column contains the authors and the year of their publications, in which the mentioned topics are addressed. Moreover, the references in these tables are also listed in reverse chronological order. The second column in Table 3 includes, in some cases, specific details of the data source used by the authors. Table 8 includes three columns: method, software, and authors.

## 3 Results

The gross number of articles identified using the search query were 29, having two matching references in Clarivate Analytics and Scopus/Elsevier: Kurnianto et al., (2019) and Eidsvig (2014). Thus, eventually, we identified 27 references. Despite the precise search query, 11 references were discarded due to reasons explained as follows. In chronological order, the first reference discarded was Papathoma-Kohle et al., (2019) because they use variables in the physical dimension, rather than socio-economic one. Two references from Yuan et al., (2019a, b) were identified by the search query as using the same method for the spatial assessment of SEV; so, we decided to select only one of them. Zhang and Huang (2018) address the topic of SV but not its spatial assessment, while Shen et al. (2018) focused on calculating the impact of disasters, rather than estimating SEV. The paper written by Goncalves, M., & Vizintim, M. F. B. (2017) was written in Portuguese, which none of the authors is proficient. Postiglione et al., (2016) promote a culture of seismic risk prevention, rather than to estimate SEV due to earthquakes. Alcántara-Ayala and Oliver-Smith (2014) present the activities undertaken by the ICL Latin -American network (ICL LAB) related to capacity building to reduce risk due to landslides, with no specific emphasis on SEV. Khazai et al., (2014), in their book chapter, concentrate on modelling shelter needs and health impacts caused by earthquakes. Vilches et al. (2014) evaluate the socio-environmental effects of the 27/10/2010 tsunami in Chile, considering the SEV among other aspects, but they do not make use of any spatial variable, indicator, or index, which is similar to the vulnerability assessment relating

to a tsunami in the Town of Tirua (Chile) undertaken by Jaque Castillo et al.,(2013). Six references from the previous search query carried out in 2018, and not identified in the refined search query, were included in the list given their relevance due to the geohazards and spatial variables, indicators, and indexes that they address. The 24 references finally reviewed are listed in Table 2.

| AUTHOR | YEAR | RESEARCH OBJECTIVE | HAZARD | COUNTRY |
|---|---|---|---|---|
| Aksha, S. K., Resler, L. M., Juran, L., & Carstensen, L. W. | 2020 | To introduce a model for spatial multi-hazards risk assessment applied to Dharan, Nepal | Earthquakes, floods and landslides | Nepal |
| Kurnianto, F. A., Ikhsan, F. A., Apriyanto, B., & Nurdin, E. A. (2019) | 2019 | To assess the level of vulnerability for an earthquake disaster in Lembang district, an area in West Java that includes the Bandung basin | Earthquakes | Indonesia |
| Muir, J. A., Cope, M. R., Angeningsih, L. R., Jackson, J. E., & Brown, R. B. | 2019 | To explore whether return migration, compared to other migration options, results in superior improvements to mental health in the context of disasters | Volcanic eruptions | Indonesia |
| Rezaei-Malek, M., Torabi, S. A., & Tavakkoli-Moghaddam, R. | 2019 | To prioritize disaster-prone areas that are known as potential demand points (PDPs) given their vulnerability under large-scale earthquakes | Earthquakes | Iran |
| Yuan, H. H., Gao, X. L., & Qi, W. (2019) | 2019 | To provide high spatial-temporal resolution information on vulnerable populations and population vulnerability using dasymetric population mapping with vulnerability index | Earthquakes | China |
| Alizadeh, M., Alizadeh, E., Kotenaee, S. A., Shahabi, H., Pour, A. B., Panahi, M., . . . Saro, L. | 2018 | To apply an artificial neural network (ANN) and geographic information system (GIS) for estimating the social vulnerability to earthquakes in the Tabriz city, Iran | Earthquakes | Iran |

| AUTHOR | YEAR | RESEARCH OBJECTIVE | HAZARD | COUNTRY |
|---|---|---|---|---|
| Qasim, S., Qasim, M., Shrestha, R. P., & Khan, A. N. | 2018 | To define the socio-economic determinants of landslide risk perception in Murree hills of Pakistan | Landslides | Pakistan |
| Ponce-Pacheco, A. B., & Novelo-Casanova, D. A. | 2018 | To estimate the levels of vulnerability and risk to floods, earthquakes and subsidence of Valle de Chalco Solidaridad (VCS) in Mexico | Earthquakes, floods and subsidence | Mexico |
| Armaş, I., Toma-Danila, D., Ionescu, R., & Gavriş, A. | 2017 | To develop an overall vulnerability index to seismic hazard based on a spatial approach applied to Bucharest, Romania | Earthquakes | Romania |
| Bereitschaft, B. | 2017 | To explore inequity in neighbourhood walkability at the micro-scale level related to social vulnerability in terms of imageability, enclosure, human scale, transparency, complexity, tidiness, and safety in Pittsburgh Streetscapes | Not walkability | USA |
| Gautam, D. | 2017 | To investigates social vulnerability to natural hazards in Nepal at district level | Droughts, earthquakes, epidemics floods and landslides, | Nepal |
| Chen, Y. | 2016 | To develop a set of valid and reliable indicators to evaluate the regional land subsidence disaster vulnerability in the Xixi-Chengnan area, in China | Landslides | China |
| Garcia, R. A. C., Oliveira, S. C., & Zezere, J. L. | 2016 | To apply dasymetric cartography to improving population spatial resolution and to assess the potentially exposed population over large areas to deep rotational landslides and compare the results with those obtained with basic census units as the data source | Landslides | Portugal |
| Maharani, Y. N., Lee, S., & Ki, S. J. | 2016 | To propose the use of Self-Organizing Maps (SOM) approach to conducting the social | Volcanic eruptions | Indonesia |

| AUTHOR | YEAR | RESEARCH OBJECTIVE | HAZARD | COUNTRY |
|---|---|---|---|---|
| | | vulnerability assessment around the Merapi volcano | | |
| Castro, C. P., Ibarra, I., Lukas, M., Ortiz, J., & Sarmiento, J. P. | 2015 | To assess the social vulnerability of informal settlements in Iquique and Puerto Montt in Chile | Earthquakes, floods, landslides and Tsunami | Chile |
| Ley-García, J., Denegri de Dios, F. M., & Ortega Villa, L. M. | 2015 | The aim is to identify visibility, invisibility and amplification of hazardscape perception in the city of Mexicali, Baja California, Mexico | Earthquake Landslide Tsunami Volcano Cyclone Thunderstorm Heavy rainfall Flood hail Snow-freeze Strong wind Drought Cold wave Heat wave | Mexico |
| Eidsvig, U. M. K., McLean, A., Vangelsten, B. V., Kalsnes, B., Ciurean, R. L., Argyroudis, S., . . . Kaiser, G. | 2014 | To propose a methodology to estimate socio-economic vulnerability to landslides at the local to regional scale using an indicator-based model | Landslides | Andorra, France, Greece, Norway, and Romania |
| Toké, N. A., Boone, C. G., & Arrowsmith, J. R. | 2014 | To construct a relative social vulnerability index classification for Los Angeles to examine the social condition within regions of significant seismic hazard, including areas regulated as Alquist-Priolo (AP) Act earthquake fault zones | Earthquakes landslides and wildfires | USA |
| Walker, B. B., Taylor-Noonan, C., Tabbernor, | 2014 | To model geophysical processes and identification of socio-economically | Earthquakes | Canada |

| AUTHOR | YEAR | RESEARCH OBJECTIVE | HAZARD | COUNTRY |
|---|---|---|---|---|
| A., McKinnon, T. B., Bal, H., Bradley, D., . . . Clague, J. J. | | disadvantaged populations in Victoria, British Columbia | | |
| Alcorn, R., Panter, K. S., & Gorsevski, P. V. | 2013 | To evaluate the spatial impact of a possible future eruption using a GIS-based volcanic hazard tool and to assess the social and economic vulnerabilities of the area at risk | Volcanic eruption | USA |
| Aubrecht, C., Özceylan, D., Steinnocher, K., & Freire, S. | 2013 | To review available multi-level geospatial information and modelling approaches from local to global scales that could serve practitioners and researchers in disaster-related zones | Tsunami, floods | Austria Portugal Turkey USA |
| Zebardast, E. | 2013 | To develop a model that combines hybrid factor analysis and analytic network process (F'ANP) for constructing a composite social vulnerability index (SOVI) | Earthquakes | Iran |
| Hizbaron, D. R., Baiquni, M., Sartohadi, J., & Rijanta, R. | 2012 | To assess urban vulnerability due to seismic hazard using a risk based spatial plan | Earthquakes | Indonesia |
| Zeng, J., Zhu, Z. Y., Zhang, J. L., Ouyang, T. P., Qiu, S. F., Zou, Y., & Zeng, T. | 2012 | To introduce a new method to assess social vulnerability for county-scale regions using population density, based on land use | Landslides | China |

Table 2. Articles identified and selected by the systematic review.

The most recurrent geohazards addressed among the selected papers are earthquakes, followed by landslides, volcanic eruptions, tsunamis, and subsidence, detailed information about the number of literature references that tackle each hazard is depicted in Figure 2. None of the references deals with soil erosion, nor land degradation. Case study areas selected from this set of papers are frequently located in Indonesia, China, Iran, and the USA, detailed information about the number of literature references that has case study areas on these countries can be appreciated in Figure 3. From the set of selected papers, the most common sources of data are the population census, followed by satellite images, field observations, disaster databases, surveys,

aerial photographs, and land use and land cover (LULC) maps. Other authors used high definition (HD) videos, orthophotos, photographs, landslide susceptibility maps, and volunteered geographic information (VGI). The complete set of data sources identified in this systematic review are listed in Table 3.

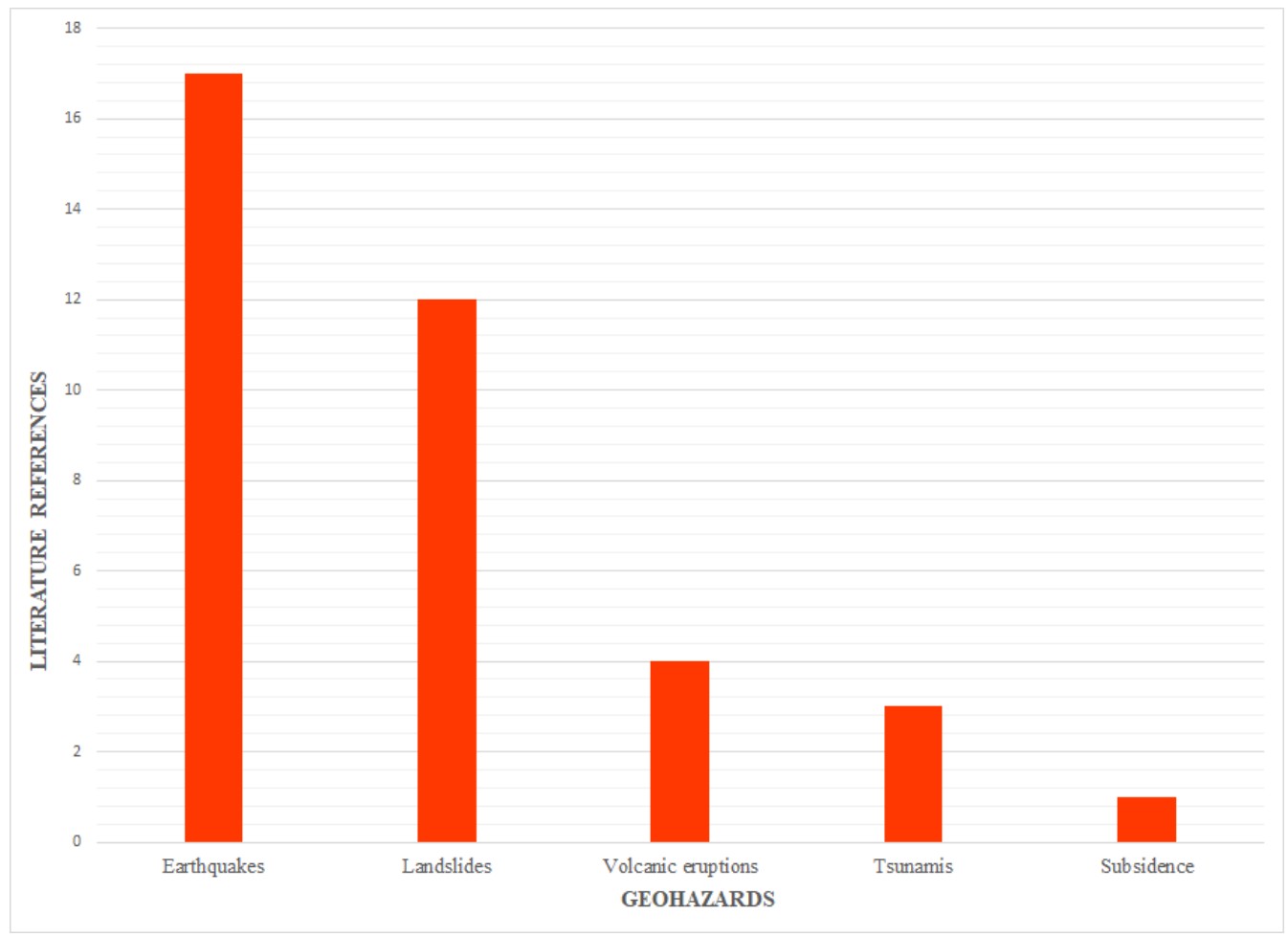

5   Figure 2. Number of literature references in the systematic review that addresses each geohazard.

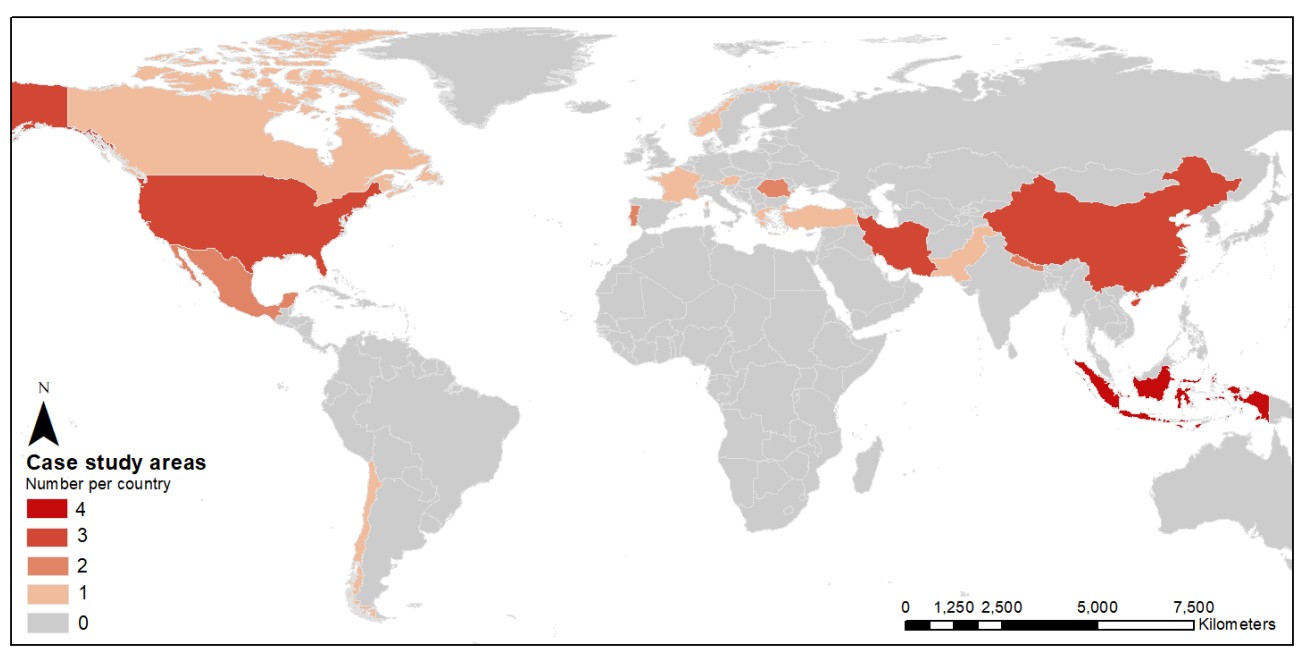

Figure 3. Number of study areas per country addressed in the references identified through the systematic literature review.

| DATA SOURCES | AUTHORS |
|---|---|
| Census data | Aksha, S. K., Resler, L. M., Juran, L., & Carstensen, L. W. (2020) |
| Nepal census | |
| | Ponce-Pacheco, A. B., & Novelo-Casanova, D. A. (2018) |
| City office of Dharan | Aksha, S. K., Resler, L. M., Juran, L., & Carstensen, L. W. (2020) |
| National Institute of Statistics and Geography | Ponce-Pacheco, A. B., & Novelo-Casanova, D. A. (2018) |
| Municipal Government of Valle de Chalco Solidaridad | Ponce-Pacheco, A. B., & Novelo-Casanova, D. A. (2018) |
| Secretariat of Social Development of Mexico | Ponce-Pacheco, A. B., & Novelo-Casanova, D. A. (2018) |
| CBS 2011 Census | Gautam, D. (2017) |
| Xishan and Huishan Statistical Yearbook 2008 | Chen, Y. (2016) |

| | DATA SOURCES | AUTHORS |
|---|---|---|
| | Population and Housing Census 2010 | Lin, W.-Y., & Hung, C.-T. (2016) |
| | National Census 2011 | Garcia, R. A. C., Oliveira, S. C., & Zezere, J. L. (2016) |
| | Statistics of Sleman Regency https://slemankab.bps.go.id/ | Maharani, Y. N., Lee, S., & Ki, S. J. (2016) |
| | National census of population and VI of housing | Castro, C. P., Ibarra, I., Lukas, M., Ortiz, J., & Sarmiento, J. P. (2015). |
| | 2000 U.S. Census Bureau | Toké, N. A., Boone, C. G., & Arrowsmith, J. R. (2014) |
| | Statistical Office of Baden-Wuerttemberg | Khazai, B., Merz, M., Schulz, C., & Borst, D. (2013) |
| | Regional Planning Board | Hizbaron, D. R., Baiquni, M., Sartohadi, J., & Rijanta, R. (2012) |
| | Statistical Bureau | Hizbaron, D. R., Baiquni, M., Sartohadi, J., & Rijanta, R. (2012) |
| | Armaş, I., Toma-Danila, D., Ionescu, R., & Gavriş, A. (2017) | |
| | Garcia, R. A. C., Oliveira, S. C., & Zezere, J. L. (2016) | |
| | Walker, B. B., Taylor-Noonan, C., Tabbernor, A., McKinnon, T. B., Bal, H., Bradley, D., . . . Clague, J. J. (2014) | |
| Satellite images | WorldView-3 | Aksha, S. K., Resler, L. M., Juran, L., & Carstensen, L. W. (2020) |
| | ASTER-DEM | Aksha, S. K., Resler, L. M., Juran, L., & Carstensen, L. W. (2020) |
| | PERSIANN-CDR | Aksha, S. K., Resler, L. M., Juran, L., & Carstensen, L. W. (2020) |
| | Google Earth satelllite images | Castro, C. P., Ibarra, I., Lukas, M., Ortiz, J., & Sarmiento, J. P. (2015) |
| | GDEM-ASTER | Castro, C. P., Ibarra, I., Lukas, M., Ortiz, J., & Sarmiento, J. P. (2015) |
| | LANDSAT | Toké, N. A., Boone, C. G., & Arrowsmith, J. R. (2014) |

| | DATA SOURCES | AUTHORS |
|---|---|---|
| | LANDSAT™ | Aubrecht, C., Özceylan, D., Steinnocher, K., & Freire, S. (2013) |
| | SPOT | Zeng, J., Zhu, Z. Y., Zhang, J. L., Ouyang, T. P., Qiu, S. F., Zou, Y., & Zeng, T. (2012) |
| | IKONOS | Aubrecht, C., Özceylan, D., Steinnocher, K., & Freire, S. (2013) |
| | NDVI | Aubrecht, C., Özceylan, D., Steinnocher, K., & Freire, S. (2013) |
| Field Observations | Alizadeh, M., Alizadeh, E., Kotenaee, S. A., Shahabi, H., Pour, A. B., Panahi, M., . . . Saro, L. (2018) | |
| | Ponce-Pacheco, A. B., & Novelo-Casanova, D. A. (2018) | |
| | Garcia, R. A. C., Oliveira, S. C., & Zezere, J. L. (2016) | |
| | Castro, C. P., Ibarra, I., Lukas, M., Ortiz, J., & Sarmiento, J. P. (2015) | |
| | Hizbaron, D. R., Baiquni, M., Sartohadi, J., & Rijanta, R. (2012) | |
| Disaster Databases | Indonesian Disaster Data Information (DIBI) http://dibi.bnpb.go.id/dibi/ | Maharani, Y. N., Lee, S., & Ki, S. J. (2016) |
| | Risk Atlas of the Municipality of Mexicali 2011 | Ley-García, J., Denegri de Dios, F. M., & Ortega Villa, L. M. (2015) |
| | Desinventar Data Base | Ponce-Pacheco, A. B., & Novelo-Casanova, D. A. (2018) |
| Surveys | Muir, J. A., Cope, M. R., Angeningsih, L. R., Jackson, J. E., & Brown, R. B. (2019) | |
| | Ponce-Pacheco, A. B., & Novelo-Casanova, D. A. (2018) | |
| | Qasim, S., Qasim, M., Shrestha, R. P., & Khan, A. N. (2018). | |
| Aerial Photograph | Castro, C. P., Ibarra, I., Lukas, M., Ortiz, J., & Sarmiento, J. P. (2015) | |
| | Toké, N. A., Boone, C. G., & Arrowsmith, J. R. (2014) | |
| LULC maps | CORINE | Aubrecht, C., Özceylan, D., Steinnocher, K., & Freire, S. (2013) |
| | HR Soil sealing layer | Aubrecht, C., Özceylan, D., Steinnocher, K., & Freire, S. (2013) |
| Population datasets | GPW/GPWv4 | Aubrecht, C., Özceylan, D., Steinnocher, K., & Freire, S. (2013) |

| DATA SOURCES | AUTHORS |
|---|---|
| GRUMP | Aubrecht, C., Özceylan, D., Steinnocher, K., & Freire, S. (2013) |
| HD video | Bereitschaft, B. (2017) |
| Orthophotos | Armaş, I., Toma-Danila, D., Ionescu, R., & Gavriş, A. (2017) |
| Photographs | Bereitschaft, B. (2017) |
| Landslide susceptibility map (pixel terrain unit) | Garcia, R. A. C., Oliveira, S. C., & Zezere, J. L. (2016) |
| VGI | Aubrecht, C., Özceylan, D., Steinnocher, K., & Freire, S. (2013) |

Table 3. Data sources for the spatial assessment of socio-economic vulnerability assessments.

The most common spatial variables used for the spatial assessment of SEV between 2010 and 2020 are households without basic services (piped water connection, electricity, sewerage infrastructure, cell phone, or landline), location, critical facilities
5 (fire stations, medical emergency services, medical facilities, and hospitals), distance from faults/causative faults, precarious housing (low quality and/or precarious external walls, roofing, and floors), the total area of occupied space in the residences, and the presence of schools. The complete set of spatial variables identified in this systematic review are listed in Table 4.

| SPATIAL VARIABLES | AUTHORS |
|---|---|
| Households without piped water connection, electricity, sewerage infrastructure, cell phone or landline | Aksha, S. K., Resler, L. M., Juran, L., & Carstensen, L. W. (2020) |
| | Ponce-Pacheco, A. B., & Novelo-Casanova, D. A. (2018) |
| | Gautam, D. (2017) |
| | Castro, C. P., Ibarra, I., Lukas, M., Ortiz, J., & Sarmiento, J. P. (2015) |
| | Zebardast, E. (2013) |
| Location | Kurnianto, F. A., Ikhsan, F. A., Apriyanto, B., & Nurdin, E. A. (2019) |
| | Muir, J. A., Cope, M. R., Angeningsih, L. R., Jackson, J. E., & Brown, R. B. (2019) |
| | Qasim, S., Qasim, M., Shrestha, R. P., & Khan, A. N. (2018) |
| | Castro, C. P., Ibarra, I., Lukas, M., Ortiz, J., & Sarmiento, J. P. (2015) |

| SPATIAL VARIABLES | AUTHORS |
|---|---|
| Critical facilities (fire stations, hospitals, health services, medical emergency services, medical facilities, etc.) | Rezaei-Malek, M., Torabi, S. A., & Tavakkoli-Moghaddam, R. (2019) |
| | Ponce-Pacheco, A. B., & Novelo-Casanova, D. A. (2018) |
| | Eidsvig, U. M. K., McLean, A., Vangelsten, B. V., Kalsnes, B., Ciurean, R. L., Argyroudis, S., . . . Kaiser, G. (2014) |
| | Alcorn, R., Panter, K. S., & Gorsevski, P. V. (2013) |
| | Zeng, J., Zhu, Z. Y., Zhang, J. L., Ouyang, T. P., Qiu, S. F., Zou, Y., & Zeng, T. (2012) |
| Distance from faults/ causative faults | Rezaei-Malek, M., Torabi, S. A., & Tavakkoli-Moghaddam, R. (2019) |
| | Hizbaron, D. R., Baiquni, M., Sartohadi, J., & Rijanta, R. (2012) |
| Household with low quality and/or precarious external walls, roofing and floors | Aksha, S. K., Resler, L. M., Juran, L., & Carstensen, L. W. (2020) |
| | Castro, C. P., Ibarra, I., Lukas, M., Ortiz, J., & Sarmiento, J. P. (2015) |
| Total area of occupied space in the residences | Armaş, I., Toma-Danila, D., Ionescu, R., & Gavriş, A. (2017) |
| | Zeng, J., Zhu, Z. Y., Zhang, J. L., Ouyang, T. P., Qiu, S. F., Zou, Y., & Zeng, T. (2012) |
| Schools | Alcorn, R., Panter, K. S., & Gorsevski, P. V. (2013) |
| | Zeng, J., Zhu, Z. Y., Zhang, J. L., Ouyang, T. P., Qiu, S. F., Zou, Y., & Zeng, T. (2012) |
| Families occupying rented houses | Aksha, S. K., Resler, L. M., Juran, L., & Carstensen, L. W. (2020) |
| Households per housing unit | Zebardast, E. (2013) |
| Households with >1 family | Aksha, S. K., Resler, L. M., Juran, L., & Carstensen, L. W. (2020) |
| City blocks | Yuan, H. H., Gao, X. L., & Qi, W. (2019) |
| Displaced, moved home, in transition, moved on | Muir, J. A., Cope, M. R., Angeningsih, L. R., Jackson, J. E., & Brown, R. B. (2019) |
| Distance to volcanoes | Kurnianto, F. A., Ikhsan, F. A., Apriyanto, B., & Nurdin, E. A. (2019) |
| Availability of evacuation roads | Ponce-Pacheco, A. B., & Novelo-Casanova, D. A. (2018) |
| Active uses/occupied storefronts | Bereitschaft, B. (2017) |
| Building color & design variety | Bereitschaft, B. (2017) |
| Building height & setback | Bereitschaft, B. (2017) |

| SPATIAL VARIABLES | AUTHORS |
| --- | --- |
| Building identifier variety | Bereitschaft, B. (2017) |
| Business type variety | Bereitschaft, B. (2017) |
| Contiguous street walls | Bereitschaft, B. (2017) |
| Courtyards, squares and parks | Bereitschaft, B. (2017) |
| | Zeng, J., Zhu, Z. Y., Zhang, J. L., Ouyang, T. P., Qiu, S. F., Zou, Y., & Zeng, T. (2012) |
| Crosswalks & ped. infrastructure | Bereitschaft, B. (2017) |
| First floor windows | Bereitschaft, B. (2017) |
| Graffiti | Bereitschaft, B. (2017) |
| Healthy/maintained vegetation | Bereitschaft, B. (2017) |
| Historic buildings | Bereitschaft, B. (2017) |
| Limited sightlines | Bereitschaft, B. (2017) |
| Litter | Bereitschaft, B. (2017) |
| Noise | Bereitschaft, B. (2017) |
| Outdoor dining | Bereitschaft, B. (2017) |
| Overhangs & vegetation | Bereitschaft, B. (2017) |
| Pedestrian activity | Bereitschaft, B. (2017) |
| Place signs/identifiers | Bereitschaft, B. (2017) |
| Public art | Bereitschaft, B. (2017) |
| Road width to building height | Bereitschaft, B. (2017) |
| Sidewalk condition | Bereitschaft, B. (2017) |
| Smells | Bereitschaft, B. (2017) |
| Street furniture | Bereitschaft, B. (2017) |
| Street vendors | Bereitschaft, B. (2017) |
| Storefront/building condition | Bereitschaft, B. (2017) |
| Street performers/entertainers | Bereitschaft, B. (2017) |
| Traffic speed | Bereitschaft, B. (2017) |
| Housing occupation type/tenancy condition | Castro, C. P., Ibarra, I., Lukas, M., Ortiz, J., & Sarmiento, J. P. (2015) |
| Average household size | Toke, N. A., Boone, C. G., & Arrowsmith, J. R. (2014) |

| SPATIAL VARIABLES | AUTHORS |
| --- | --- |
| Housing type | Eidsvig, U. M. K., McLean, A., Vangelsten, B. V., Kalsnes, B., Ciurean, R. L., Argyroudis, S., . . . Kaiser, G. (2014) |
| Percentage of households with public assistance | Toke, N. A., Boone, C. G., & Arrowsmith, J. R. (2014) |
| Percent of workers with a long commute | Toke, N. A., Boone, C. G., & Arrowsmith, J. R. (2014) |
| Travel barriers to the trauma centres | Walker, B. B., Taylor-Noonan, C., Tabbernor, A., McKinnon, T. B., Bal, H., Bradley, D., . . . Clague, J. J. (2014) |
| Travel distance to trauma centres | Walker, B. B., Taylor-Noonan, C., Tabbernor, A., McKinnon, T. B., Bal, H., Bradley, D., . . . Clague, J. J. (2014) |
| Travel time to trauma centres | Walker, B. B., Taylor-Noonan, C., Tabbernor, A., McKinnon, T. B., Bal, H., Bradley, D., . . . Clague, J. J. (2014) |
| Walking time to trauma centres | Walker, B. B., Taylor-Noonan, C., Tabbernor, A., McKinnon, T. B., Bal, H., Bradley, D., . . . Clague, J. J. (2014) |
| Land use | Alcorn, R., Panter, K. S., & Gorsevski, P. V. (2013) |
| Housing with bathroom | Zebardast, E. (2013) |
| Housing with kitchen | Zebardast, E. (2013) |
| Migration status | Muir, J. A., Cope, M. R., Angeningsih, L. R., Jackson, J. E., & Brown, R. B. (2019) |
| Road type | Alcorn, R., Panter, K. S., & Gorsevski, P. V. (2013) |
| Spatial distribution of cell phone subscribers | Aubrecht, C., Özceylan, D., Steinnocher, K., & Freire, S. (2013) |
| Distance to hospital | Zeng, J., Zhu, Z. Y., Zhang, J. L., Ouyang, T. P., Qiu, S. F., Zou, Y., & Zeng, T. (2012) |
| Distance to road network | Hizbaron, D. R., Baiquni, M., Sartohadi, J., & Rijanta, R. (2012) |
| Distance to trauma centres | Walker, B. B., Taylor-Noonan, C., Tabbernor, A., McKinnon, T. B., Bal, H., Bradley, D., . . . Clague, J. J. (2014) |
| Distribution of urban greenspace | Toké, N. A., Boone, C. G., & Arrowsmith, J. R. (2014) |

| SPATIAL VARIABLES | AUTHORS |
|---|---|
| Industry land, Office land and commercial and residential land | Zeng, J., Zhu, Z. Y., Zhang, J. L., Ouyang, T. P., Qiu, S. F., Zou, Y., & Zeng, T. (2012) |
| Population dependent on the land for the primary source of income | Eidsvig, U. M. K., McLean, A., Vangelsten, B. V., Kalsnes, B., Ciurean, R. L., Argyroudis, S., . . . Kaiser, G. (2014). |
| Road network | Zeng, J., Zhu, Z. Y., Zhang, J. L., Ouyang, T. P., Qiu, S. F., Zou, Y., & Zeng, T. (2012) |

Table 4. Spatial variables for socio-economic vulnerability assessments.

Population density, housing density, hospital beds per 1,000 people, and living space per person are the most frequent spatial indicators of SEV. Global Moran's I and local indicators of spatial association (LISA), which are traditional indicators in the spatial assessment, were also identified in this systematic research. We also found indicators such the access to environmental amenities and medical facilities, mobility, employed/unemployed density, and literate people density among others. The complete set of spatial indicators identified in this systematic review are listed in Table 5.

.

| SPATIAL INDICATORS | AUTHORS |
|---|---|
| Population density (women/men density) | Alizadeh, M., Alizadeh, E., Kotenaee, S. A., Shahabi, H., Pour, A. B., Panahi, M., . . . Saro, L. (2018) |
| | Kurnianto, F. A., Ikhsan, F. A., Apriyanto, B., & Nurdin, E. A. (2019) |
| | Yuan, H. H., Gao, X. L., & Qi, W. (2019) |
| | Alizadeh, M., Alizadeh, E., Kotenaee, S. A., Shahabi, H., Pour, A. B., Panahi, M., . . . Saro, L. (2018) |
| | Armaş, I., Toma-Danila, D., Ionescu, R., & Gavriş, A. (2017) |
| | Chen, Y. (2016) |
| | Maharani, Y. N., Lee, S., & Ki, S. J. (2016) |
| | Eidsvig, U. M. K., McLean, A., Vangelsten, B. V., Kalsnes, B., Ciurean, R. L., Argyroudis, S., . . . Kaiser, G. (2014) |
| | Toké, N. A., Boone, C. G., & Arrowsmith, J. R. (2014) |
| | Hizbaron, D. R., Baiquni, M., Sartohadi, J., & Rijanta, R. (2012) |

| SPATIAL INDICATORS | AUTHORS |
|---|---|
| | Zeng, J., Zhu, Z. Y., Zhang, J. L., Ouyang, T. P., Qiu, S. F., Zou, Y., & Zeng, T. (2012) |
| Housing density | Alizadeh, M., Alizadeh, E., Kotenaee, S. A., Shahabi, H., Pour, A. B., Panahi, M., . . . Saro, L. (2018) |
| | Armaş, I., Toma-Danila, D., Ionescu, R., & Gavriş, A. (2017) |
| | Toké, N. A., Boone, C. G., & Arrowsmith, J. R. (2014) |
| Hospital beds per 1,000 people | Eidsvig, U. M. K., McLean, A., Vangelsten, B. V., Kalsnes, B., Ciurean, R. L., Argyroudis, S., . . . Kaiser, G. (2014) |
| | Zebardast, E. (2013) |
| Mobility | Yuan, H. H., Gao, X. L., & Qi, W. (2019) |
| | Bereitschaft, B. (2017) |
| Living space pp | Armaş, I., Toma-Danila, D., Ionescu, R., & Gavriş, A. (2017) |
| | Zebardast, E. (2013) |
| Degree of population aglomeration | Yuan, H. H., Gao, X. L., & Qi, W. (2019) |
| Floating population | Yuan, H. H., Gao, X. L., & Qi, W. (2019) |
| Spatial distribution | Yuan, H. H., Gao, X. L., & Qi, W. (2019) |
| Employed/ Unemployed density | Alizadeh, M., Alizadeh, E., Kotenaee, S. A., Shahabi, H., Pour, A. B., Panahi, M., . . . Saro, L. (2018). |
| Household overcrowding | Ponce-Pacheco, A. B., & Novelo-Casanova, D. A. (2018) |
| Literate people density | Alizadeh, M., Alizadeh, E., Kotenaee, S. A., Shahabi, H., Pour, A. B., Panahi, M., . . . Saro, L. (2018) |
| Businesses density | Bereitschaft, B. (2017) |
| Complexity | Bereitschaft, B. (2017) |
| Enclosure | Bereitschaft, B. (2017) |
| Human scale | Bereitschaft, B. (2017) |
| Imageability | Bereitschaft, B. (2017) |
| Safety & sensations | Bereitschaft, B. (2017) |
| Tidiness | Bereitschaft, B. (2017) |
| Traffic density | Bereitschaft, B. (2017) |
| Transparency | Bereitschaft, B. (2017) |
| BCU target zones | Garcia, R. A. C., Oliveira, S. C., & Zezere, J. L. (2016) |

| SPATIAL INDICATORS | AUTHORS |
| --- | --- |
| BCU population | Garcia, R. A. C., Oliveira, S. C., & Zezere, J. L. (2016) |
| Density of agricultural/industrial production | Chen, Y. (2016) |
| Farming density | Chen, Y. (2016) |
| GDP density | Chen, Y. (2016) |
| Investment density of fixed assets | Chen, Y. (2016) |
| Global Moran's I | Ley-García, J., Denegri de Dios, F. M., & Ortega Villa, L. M. (2015) |
| LISA | Ley-García, J., Denegri de Dios, F. M., & Ortega Villa, L. M. (2015) |
| Access to environmental amenities (Park space, open spaces and walkable neighborhoods) | Toké, N. A., Boone, C. G., & Arrowsmith, J. R. (2014) |
| Access to medical facilities | Walker, B. B., Taylor-Noonan, C., Tabbernor, A., McKinnon, T. B., Bal, H., Bradley, D., . . . Clague, J. J. (2014) |
| Infrastructure dependance | Toke, N. A., Boone, C. G., & Arrowsmith, J. R. (2014) |
| Walkability | Toké, N. A., Boone, C. G., & Arrowsmith, J. R. (2014) |

Table 5. Spatial indicators for socio-economic vulnerability assessments.

Results extracted from the literature indicate that that the most common methods in the last 10 years for the reduction of variables is principal component analysis (PCA) and for indicators weighting is Analytic Hierarchy Process (AHP). The use of artificial neural networks (ANN) has been gaining ground in the last 10 years as a method for the spatial assessment of SEV. Other methods include dasymetric population mapping, factor analysis (FA), ordinal logistic regression (OLR), spatial multi-criteria evaluation (SMCE), and analytic network process (ANP). We also found hybrid methods that combine FA and ANP known as F'ANP, and others that combine fuzzy numbers with ANP, DEMATEL and PROMETHEE II (F-ADP). Other methods were simpler, such as an overlay analysis. The complete set of methods used by authors and identified in this systematic review is listed in Table 6.

| METHODS | AUTHORS |
| --- | --- |
| PCA | Aksha, S. K., Resler, L. M., Juran, L., & Carstensen, L. W. (2020) |
| | Armaş, I., Toma-Danila, D., Ionescu, R., & Gavriş, A. (2017) |
| | Maharani, Y. N., Lee, S., & Ki, S. J. (2016) |
| | Toké, N. A., Boone, C. G., & Arrowsmith, J. R. (2014) |
| | Alcorn, R., Panter, K. S., & Gorsevski, P. V. (2013) |
| AHP | Aksha, S. K., Resler, L. M., Juran, L., & Carstensen, L. W. (2020) |
| | Alizadeh, M., Alizadeh, E., Kotenaee, S. A., Shahabi, H., Pour, A. B., Panahi, M., . . . Saro, L. (2018) |
| | Armaş, I., Toma-Danila, D., Ionescu, R., & Gavriş, A. (2017) |
| | Walker, B. B., Taylor-Noonan, C., Tabbernor, A., McKinnon, T. B., Bal, H., Bradley, D., . . . Clague, J. J. (2014) |
| ANN | Aksha, S. K., Resler, L. M., Juran, L., & Carstensen, L. W. (2020) |
| | Alizadeh, M., Alizadeh, E., Kotenaee, S. A., Shahabi, H., Pour, A. B., Panahi, M., . . . Saro, L. (2018) |
| | Maharani, Y. N., Lee, S., & Ki, S. J. (2016) |
| Dasymetric population mapping | Yuan, H. H., Gao, X. L., & Qi, W. (2019) |
| | Garcia, R. A. C., Oliveira, S. C., & Zezere, J. L. (2016) |
| FA | Castro, C. P., Ibarra, I., Lukas, M., Ortiz, J., & Sarmiento, J. P. (2015) |
| | Zebardast, E. (2013) |
| MCE | Walker, B. B., Taylor-Noonan, C., Tabbernor, A., McKinnon, T. B., Bal, H., Bradley, D., . . . Clague, J. J. (2014) |
| | Alcorn, R., Panter, K. S., & Gorsevski, P. V. (2013) |
| SMCE | Armaş, I., Toma-Danila, D., Ionescu, R., & Gavriş, A. (2017) |
| | Hizbaron, D. R., Baiquni, M., Sartohadi, J., & Rijanta, R. (2012) |
| F-ADP | Rezaei-Malek, M., Torabi, S. A., & Tavakkoli-Moghaddam, R. (2019) |
| OLR | Muir, J. A., Cope, M. R., Angeningsih, L. R., Jackson, J. E., & Brown, R. B. (2019) |
| Binary Logistic regression | Qasim, S., Qasim, M., Shrestha, R. P., & Khan, A. N. (2018) |
| Logical analysis method | Chen, Y. (2016) |
| Distance-based network analysis | Walker, B. B., Taylor-Noonan, C., Tabbernor, A., McKinnon, T. B., Bal, H., Bradley, D., . . . Clague, J. J. (2014) |
| Overlay analysis | Toké, N. A., Boone, C. G., & Arrowsmith, J. R. (2014) |

| METHODS | AUTHORS |
|---|---|
| F'ANP | Zebardast, E. (2013) |

Table 6. Methods applied to the spatial assessment of socio-economic vulnerability.

The Social Vulnerability Index (SoVI®) remains the benchmark for the assessment of SEV and a reference for its spatial assessment. Nevertheless, indices such as Walk Scores® (Bereitschaft, 2017a) offer a proxy for the spatial assessment of SEV

5   in a microscale urban level (street level) in 3 dimensions (3D). The complete set of spatial indexes used by authors and identified in this systematic review is listed in Table 7.

| SPATIAL INDEXES | AUTHORS |
|---|---|
| SoVI® | Aksha, S. K., Resler, L. M., Juran, L., & Carstensen, L. W. (2020) |
| | Alcorn, R., Panter, K. S., & Gorsevski, P. V. (2013) |
| | Zebardast, E. (2013) |
| Population vulnerability Indexing | Yuan, H. H., Gao, X. L., & Qi, W. (2019) |
| Walk Scores® | Bereitschaft, B. (2017) |
| LA-SoVIC | Toké, N. A., Boone, C. G., & Arrowsmith, J. R. (2014) |

Table 7. Spatial indexes for socio-economic vulnerability assessments.

The tools to carry out the spatial assessment of SEV were selected according to the identified spatial variable and indicators,

10   the method used, and the indexes used, adapted, or developed. The most frequent tool for the spatial assessment of SEV is GIS, followed by statistical analyses undertaken in the statistical package for the social sciences (SPSS), remote sensing (RS) using the environment for visualizing images (ENVI), programming languages, and interactive databases such as the retrieval of data for small Areas by microcomputer (REDATAM)(CELADE, 2015). The complete list of tools used by the authors selected is found in Table 8.

| METHOD | SOFTWARE | AUTHORS |
|---|---|---|
| GIS | ArcGIS | Aksha, S. K., Resler, L. M., Juran, L., & Carstensen, L. W. (2020). |
| | | Yuan, H. H., Gao, X. L., & Qi, W. (2019) |

| METHOD | SOFTWARE | AUTHORS |
|---|---|---|
| | | Alizadeh, M., Alizadeh, E., Kotenaee, S. A., Shahabi, H., Pour, A. B., Panahi, M., . . . Saro, L. (2018) |
| | | Gautam, D. (2017) |
| | | Castro, C. P., Ibarra, I., Lukas, M., Ortiz, J., & Sarmiento, J. P. (2015) |
| | IDRISI | Alizadeh, M., Alizadeh, E., Kotenaee, S. A., Shahabi, H., Pour, A. B., Panahi, M., . . . Saro, L. (2018) |
| | ILWIS | Armaş, I., Toma-Danila, D., Ionescu, R., & Gavriş, A. (2017) |
| | GeoDa Version 16.6 | Ley-García, J., Denegri de Dios, F. M., & Ortega Villa, L. M. (2015) |
| | Not specified | Ponce-Pacheco, A. B., & Novelo-Casanova, D. A. (2018) |
| | | Eidsvig, U. M. K., McLean, A., Vangelsten, B. V., Kalsnes, B., Ciurean, R. L., Argyroudis, S., . . . Kaiser, G. (2014) |
| | | Toke, N. A., Boone, C. G., & Arrowsmith, J. R. (2014) |
| | | Walker, B. B., Taylor-Noonan, C., Tabbernor, A., McKinnon, T., Bal, H., Bradley, D., . . . Clague, J. J. (2014) |
| | | Alcorn, R., Panter, K. S., & Gorsevski, P. V. (2013) |
| | | Hizbaron, D. R., Baiquni, M., Sartohadi, J., & Rijanta, R. (2012) |
| Statistical Analysis | SPSS 22.0 | Aksha, S. K., Resler, L. M., Juran, L., & Carstensen, L. W. (2020) |
| | SPSS 16.0 | Qasim, S., Qasim, M., Shrestha, R. P., & Khan, A. N. (2018) |
| | SPPS | Maharani, Y. N., Lee, S., & Ki, S. J. (2016) |
| | | Castro, C. P., Ibarra, I., Lukas, M., Ortiz, J., & Sarmiento, J. P. (2015) |
| RS | ENVI | Alizadeh, M., Alizadeh, E., Kotenaee, S. A., Shahabi, H., Pour, A. B., Panahi, M., . . . Saro, L. (2018) |
| Programming language | MATLAB | Maharani, Y. N., Lee, S., & Ki, S. J. (2016) |
| Database | Redatam V5.0 | Castro, C. P., Ibarra, I., Lukas, M., Ortiz, J., & Sarmiento, J. P. (2015) |

Table 8. Tools for socio-economic vulnerability assessments.

## 4 Discussion

For the purpose of the systematic review, we found that the Clarivate Analytics database more accurately identified the references for this systematic review, and it is more user-friendly than other databases. The lack of articles that tackle exogenic geohazards can be explained by the fact that we excluded from the search query words such as "climate change" OR "ecological" OR "drought", which are indirectly related to these phenomena. Nevertheless, considering that these geohazards usually takes place in rural, rather than urban environments, they are not relevant for this research.

The literature references identified are based on a high detailed search query to avoid bias. The query could be repeated any time and the results will be always the same, maybe additional publications from 2020 could appear on the result. However, the total number of literature references reviewed were much more than 24. Previously, based on a more general query not specifically focused on geohazards, we identified 235 literature references, from which we found 84 relevant references, 42 highly relevant references and finally 21 references were selected to be reviewed at that moment. Eventually, given their relevance, we decided to keep six of these references identified previously using the first query. In the current version, we reviewed all 29 references but eventually, we selected 18 and discarded 11 for the reasons already explained in the results section. The case study areas of the selected papers confirm the findings from Shen et al., (2018) and also ours using the previous query, relating to the USA, China, and Iran as major contributors to disaster research together with Italy, Indonesia, Germany, Turkey, England, India, and Spain in the topics of 'prediction model', 'social vulnerability' and 'landslide inventory map'. Nevertheless, the references that use Indonesia as a case study area are focused on earthquakes and volcanic eruptions, not necessarily on the tsunami hazard as was suggested by Shen et al., (2018). The reason to lead the research in those topics would be based on their degree of hazard considering that the USA, China and Indonesia are located along the Pacific Ring of Fire.

The research concentrated on the local level uses primary data collected via field observations, questionnaire surveys, or focus groups with representative members of the community to assess vulnerability (Birkmann, 2006; Khazai et al., 2017; Sarkar and Vogt, 2015), while for global or regional scales, primary data is derived from satellite images, aerial Photograph, LULC,

landslide susceptibility maps, orthophotos, or VGI. Secondary data is obtained from the population census, disaster databases, and population datasets. For applications on the regional, national, international, or worldwide scale, coarse-scale raster data on population patterns are appropriate, but for city or local scales, representation of higher spatial resolution is requested, such as fine-scale population grids which finally go to individual building level (Aubrecht et al., 2013). Census data usually presents

national data at the municipal level. Census and land databases are highly demanded by planners and disaster managers. However, there are several problems associated with using large community databases, such as scale, data decay, relevance (King, 2001), and time-constrains. Current data can easily change with the building of a new road or new houses (McLaughlin et al., 2002), and in the case of nomadic and/or geographically isolated groups, these data sets are rarely available (Béné, 2009) but they are necessary. Censuses are usually updated on an average of ten years, depending on the country, and some of the

data could be altered by political biases. The surveys require significant resources, and the thematic scope is usually very narrow. These disadvantages can explain the strong demand for population data, independent of administrative areas, making it sometimes necessary to extract data from raster representations or using dasymetric mapping (Aubrecht et al., 2013; Garcia et al., 2016; Yuan et al., 2019a). Currently, data in 3D can be also extracted from VGI, which is an alternative source of real-time information based on the concept of citizens as sensors (Cervone and Hultquist, 2018).

Satellite images are useful to collect data from global to local scales. Rapid mapping concepts are mainly applied in structural post-disaster damage assessment, relaying on earth observation data from different sensors, sometimes provided by the International Charter Space and Major Disasters (2020) (Aubrecht et al., 2013). Lidar data are a good option for the city scale. The use of satellite images as data sources in the spatial assessment of SEV has been increasing in the last ten years, which

can be explained because they offer quick, updated, and reliable data, making the satellite images currently the most effective source. One of the issues with using maps, air photos, or orthophotos as a resource is that they are not frequently updated.

The spatial variables found through this systematic review are similar to the variables identified by Meentemeyer (1989), Béné, (2009) Contreras et al. (2013), Buzai and Villerías Alarcón (2018). Based on the concept of spatial indicator of SV formulated

by Ebert et al., (2009), we consider the lack of basic services as a spatial variable of SEV because all these networks are

distributed in a specific spatial area. The lack of life-supporting infrastructure and/or infrastructure necessary for the functioning of the society such as piped water, electricity networks, sewerage infrastructure, telecommunications and road networks hampers emergency management and therefore the recovery process (Eidsvig et al., 2014). Housing quality and tenancy conditions describe the vulnerability of the population to become homeless after a disaster (Toke et al., 2014). Housing

type is an economic indicator of the economic status of individuals, communities, and nations. Thus, a house with low quality or precarious external walls located in a landslide-prone zone is usually associated with socially vulnerable communities having a negative influence on the quality of life. However, the typology of vulnerable houses depends also on the sort of landslide (Eidsvig et al., 2014). There are similar spatial variables used to produce an indicator of housing overcrowding (Ponce-Pacheco and Novelo-Casanova, 2018) such as households per housing unit (Zebardast, 2013) and households with >1

family (Aksha et al., 2020). We argue that besides spatial variables, we must also consider spatial categories in which critical and the other urban facilities must be included. These facilities are not only providers of services but are also sources of employment (Contreras et al., 2017); therefore, their presence or absence, access to, distance, travel time (Toke et al., 2014), and/or barriers (Walker et al., 2014) to reaching them highly influence the degree of spatial SEV of a community. Bereitschaft, (2017a) proposes innovative spatial variables of SEV at microscale urban level in 3D such as historic buildings, parks, place

signs/identifiers, contiguous street wall, limited sightlines, street furniture, street vendors, first-floor windows, active uses/occupied storefronts, pedestrian activity, business type variety, crosswalks & pedestrian infrastructure, sidewalk condition and storefront/building condition. We also identify other spatial variables that are different to more traditional ones such as distance from faults (Hizbaron et al., 2012; Rezaei-Malek et al., 2019) and volcanoes (Kurnianto et al., 2019), land use (Alcorn et al., 2013), city blocks (Yuan et al., 2019a) and displacement (Muir et al., 2019) among others.

Based on the evidence found by this research, we agree with Zeng et al. (2012) that the most frequent spatial indicator in the assessment of SEV related to geohazards is population density, and it has the highest sensitivity coefficient (Yuan et al., 2019a). According to Kurnianto et al., (2019), high population density is the factor that contributes most to the high SV, and it is usually linked to high population growth, which increases the SEV given the rise in the exposure of population and business.

The reason, according to Gu et al. (2018), is that population density reveals the human resources of a neighbourhood and the

relief resources that could be required during a disaster. This is a key factor in large case study areas where different kinds of occupation can take place (urban, rural); therefore, important differences in population density are expected to be found. Disadvantaged population tends to live in denser neighbourhoods with more crowded parks and other recreational facilities (Sister et al., 2009; Toke et al., 2014; Wolch et al., 2005) and low levels of walkability (Bereitschaft, 2017a) that exacerbate the vulnerability making an evacuation difficult (Cutter et al., 2003) after an earthquake, tsunami, volcanic eruption, or landslide. It is also more difficult in such areas to find spaces to install temporary shelters near their households or areas for providing care after an emergency (Cutter et al., 2003). The density of the built environment is especially important in the case of seismic events (Toke et al., 2014). Innovative spatial indicators such as employed density, unemployed density, and literacy people density were proposed by Alizadeh et al. (2018). The importance of such fine-scale data and the temporal variations (daytime and night-time) for accurately estimating SV was highlighted by Yuan et al., (2019a), proposing the indicator: 'floating population'. The consideration of the spatial and temporal dimension in the estimation of population exposure is a fundamental aspect of accurate catastrophe loss modelling, a key element for the integration of risk analysis and emergency management (Aubrecht et al., 2010), and therefore for the reduction of the SEV (Alizadeh et al., 2018). Chen (2016) proposes more spatial indicators in the economic rather than the social dimension. Ley-García et al. (2015), global Moran's I and LISA enable the identification of dependence between attributes and localisations. As a result, these indicators are useful to determine whether the spatial distribution of elements influences the behaviour of a particular variable. The summary measure of autocorrelation in the territory is undertaken with global Moran's I, while the autocorrelation of the spatial units included in the territory is measured using LISA. Cutter and Finch (2008) also previously utilised global Moran's I and LISA to identify local variability and cluster similarity of low and SV. Besides the SoVI® and FA, Zhou et al. (2014) utilise exploratory spatial data analysis (ESDA) to identify the spatio-temporal patterns of SV based on the constructed SoVI® for each county in China. These authors used global and local Moran's I or LISA as ESDA to determine the spatial autocorrelation among counties and identify the similarity and/or dissimilarity in the clustering of SV.

Accessibility as a spatial indicator is defined as the ability to contact and interact with places of economic or social opportunities (Deichmann, 1997). Goodall (1987) notes that accessibility is the ease to reach a location from another location, and this concept is also related to opportunities for attention (Aubrecht et al., 2013) in the case of, for example, hospitals and/or

trauma centres, accessibility is reduced by distance (Hizbaron et al., 2012; Zeng et al., 2012) increasing SEV level of the communities located far from these healthcare facilities. Besides the common spatial variables, indicators and indexes in 2D, there are also spatial indicators and indexes that include a 3D component, such as imageability, enclosure, human scale, transparency, complexity, safety & sensations and tidiness (Bereitschaft, 2017a) satisfaction with the neighbourhood (Barata

et al., 2011), and residential condition (de la Torre and de Riccitelli, 2017) that could be applied to the spatial assessment of SEV. Authors such as Yuan et al (2020) and Muir et al (2019) consider the spatial indicators of mobility and migration respectively in the framework of geohazards, being migration a topic mainly addressed by authors in the climate change community e.g. Nakayama et al.,(2019), Naugle et al.,(2019), van der Geest et al., (2020), Ayeb-Karlsson et al.,(2020) and others.

This systematic review identified the versatility of ANN, which can be either used to extract monthly rainfall data (Aksha et al., 2020), for deriving social vulnerability maps (SVM) (Alizadeh et al., 2018) or to train the self-organized map (SOM) algorithm cluster method (Maharani et al., 2016). The use of dasymetric population mapping not limited to administrative boundaries, even going to block-level to increase the spatial resolution of the population exposure analysis (Garcia et al., 2016)

and additionally by including the temporal dimension with its day-night variability, enables improving the accuracy of the spatial assessments of SEV (Yuan et al., 2019a). Factor analysis (FA) is used by Castro et al. (2015) to establish the level of SEV and by Zebardast (2013) to extract primary dimensions and variables of SEV. Alcorn (2013) applied MCE to assess economic vulnerability using four significant factors: population, infrastructure, land use, and economic production. SMCE is applied by Armaş et al., (2017) to integrate social, education, housing, and social dependence vulnerability dimensions and by

Hizbaron (2012) to develop deterministic SV scenarios. Zebardast (2013) enters the variables of SEV into a network model in an analytic network process (ANP) to rank the importance of each variable to complete the F'ANP method. This method is focused on developing a composite social vulnerability index (SOVI). Binary logistic regression was the statistical method applied by Qasim et al. (2018) to identify the determinants of landslide risk perception, location being one of them. Walker et al., (2014) present a multi-criteria evaluation (MCE) model that incorporates access to healthcare facilities using GIS to identify

and rank residential areas in Victoria, British Columbia. The integration of the concept of uncertainty into ANP using fuzzy

numbers (F-ANP) is combined by Rezaei-Malek et al., (2019) with fuzzy DEMATEL (F-DEMATEL) to deal with the interdependency among a set of criteria and fuzzy PROMETHEE II (F- PROMETHEE II) to control the criteria weights, the complete method is denominated fuzzy ANP DEMATEL PROMETHEE II (F-ADP). Ordinal logistic regression (OLR) is used by Muir et al., (2019) to predict the mental health condition of people displaced by series of volcanic eruptions in Merapi,

Indonesia, according to their migration status (displaced, moved home, in transition, and moved on), which implies a spatial component. Geological experience and logical analysis method were used by Chen (2016) to select indicators. Toke et al.,(2014) undertake an overlay analysis to identify the census block groups that intersect zones with an extreme ground shaking hazard.

Aksha et al., (2020) utilized the SoVI® to map the vulnerability levels in the study site with a multi-hazard map to produce a

total risk map. Alcorn et al. (2013) used an improved version of the same index but specifically adapted it to the variability in SEV in the case study area that was focused on census-designated places (CDPs) on a small scale. The population vulnerability indexing developed by Yuan et al., (2019a) considered most of the indicators available in the literature already identified by the SoVI®, but they adapted their index to the Chinese society, where according to the authors, race and ethnicity are not relevant indicators and rural-to-urban migrants are floating population with unequal access to public services and therefore a

vulnerable population. Bereitschaft (2017a) explores the exiting inequities in the walkability of urban environments among neighbourhoods with low and high SEV using the Walk Scores®. This index could be used as a proxy spatial index of SEV in 3D at microscale urban level. The author found that neighbourhoods with high SV had fewer windows and less transparent storefronts, less continuous street walls, less well-maintained infrastructure, fewer business and generally less complexity than in neighbourhoods with low SV. Toké et al (2014), build upon the SoVI® to create their own SV indexes that incorporate the

spatial dimension. According to the LA-SoVIC developed by Toket et al. (2014), SV is highly linked to the normalised difference vegetation index (NDVI) as a proxy for urban green space. Green areas are usually located in areas with lower SEV (Stow et al., 2007), and have also been recognised for their health benefits (Bedimo-Rung et al., 2005). Physical characteristics of green areas, such as attractive scenery, motivates people to stay and visit an area (Kurnianto et al., 2019), resulting in increased social control and reduced SEV.

It has been always difficult to quantify SV; hence, it is absent from post-disaster cost/loss estimation reports (Schmidtlein et al., 2008; Zhou et al., 2014). The use of spatial variables, indicators, and indexes will bridge the gap of integrating physical vulnerability and SV to achieve a holistic risk assessment. Davidson (1997) provides the first attempt to create an integrated risk assessment framework. Later, Carreño, Cardona, & Barbat, (2007) developed a risk index obtained by multiplying the

physical risk index by an impact factor, which is, in fact, an aggravating coefficient consisting of socio-economic variables; nevertheless, in applying this method, the outcome will be similar to the assessment of physical vulnerability, without showing the contribution of SV to the assessment of integrated risk. Schmidtlein, Shafer, Berry and Cutter (2011) tested the link between SV and earthquake losses. The authors found that physical parameters related to hazard, such as distance from the epicentre and peak ground acceleration, were more significant in predicting impacts than SV. Nevertheless, the same authors established

that SV is a significant predictor of earthquake losses when accounting for wealth (dollar losses per average income as the dependent variable). The previous finding reveals that those areas with higher levels of SV experience a greater relative impact than areas with lower degrees of SV.

Geospatial information systems are broadly utilised by several authors to collect and, process data, and map the SEV. GIS has been enabling researchers to have either large study regions, or equivalently, data sets at much finer spatial resolution (Unwin,

1996), for example, a comprehensive overview of the use of accessibility indicators in GIS was already provided by Deichmann (1997). Each author uses different versions of ArcGIS, which is the most widespread software used in GIS. The IDRISI software is utilised by Alizadeh et al. (2018) to generate a Social Vulnerability Map (SVM). Armaş et al., (2017) applied a pairwise comparative method in the AHP implemented in the SMCE module of the Integrated Land and Water Information System (IlWIS) software. GeoDa, an open-source software, focused on methods for spatial data and has been used by authors

who address the topic of spatial association (Gu et al., 2018; Ley-García et al., 2015). The aforementioned is an RS and GIS software, on which the robustness of the results from Armaş et al. (2017) was also tested, with a sensitivity analysis performed in the DEFINITE toolbox implemented in IlWIS. The MATLAB computation environment was used by Maharani et al. (2016) to develop the SOM toolbox. Sherly et al. (2015) also use MATLAB to perform multivariate data analyses, such as PCA and Data Envelopment Analysis (DEA). REDATAM used as a source of data by Castro et al. (2015), is an interactive hierarchical

database that contains microdata and/or aggregate socio-economic information from any geographical division at a national

level. This database combines data from the census, surveys and other sources, resulting in a very comprehensive and useful source of spatial and not-spatial variables for the SEV.

**5 Conclusions**

Based on the evidence, we can state that most of the spatial assessments of SEV in urban environments have been done for earthquakes and landslides and that Indonesia, China, Iran, and the USA lead the research in the spatial assessment of SEV related to geohazards in urban environments. The scale of the spatial level of assessment – namely global, continental, subcontinental, national, regional, provincial, municipal, or local – determines the type of data to be collected and the assessment approaches. Although there have been advances, census data continues to be the most frequent source of data for the SEV assessments; however, in the case of spatial assessment, satellite images are now the main data source, facilitating the inclusion of the spatial component in SEV assessments. The spatial assessment of SEV allows visualising and communicating social phenomena and components that influence the degree of vulnerability that are not visible with other methods. The lack of data availability hinders the understanding of the concept of vulnerability (Zhou et al., 2014) and that is why VGI is essential today to obtain updated information in real-time at the local scale when other data sources are not available.

Traditional spatial variables and indicators continue to be used by authors, but combined with new variables, categories, and indicators, including the temporal dimension (day-night), and assessing at the local level, can increase the accuracy of spatial assessments of SEV and reduce uncertainty on their assessment. Each method for the spatial assessment of SV is selected according to the research aim, case study area, scale to cover, reliability of data sources, spatial variables and indicators available; geohazard to address, the scope of the research, and the level of funding. Methods such as ANN are gaining ground in the assessment of SEV. Other methods such as dasymetric population mapping enable more accurate SEV assessment. Factor analysis continues to be a useful tool to define the level of SEV based on primary dimensions and variables. Multi-criteria evaluation method offers a robust decision-making technique based on flexible choice and combination in criteria (Alcorn et al., 2013). SMCE incorporates the spatial component to the MCE to integrate spatial and non-spatial data to generate maps with multiple scenarios (Hizbaron et al., 2012). Classic methods such as FA are combined with more innovative ones

such as ANP and fuzzy numbers to generate hybrid methods such as F'ANP. These new methods encourage the development of more complex hybrid methods such as F-ADP that increase the accuracy and reduce the uncertainty levels in the spatial SEV assessments. Ordinal logistic regression and binary logistic regression are useful methods to identify spatial variables as determinants of SEV. The spatial component can be also be added by simply overlapping the areas with high SEV with hazard

zones using GIS. Most authors have built upon the SoVI® developed by Cutter et al. (2003) to quantify SEV or to create their own SEV indexes, demonstrating that it remains the benchmark for the assessment of SEV and a reference for its spatial assessment, however there are new alternatives for the spatial assessment of SEV in 3D at microscale level such as Walk Scores® (Bereitschaft, 2017a) .

Geographic Information Systems, statistical analysis, RS, programming languages, and interactive databases are the tools

currently used by the scientists for the assessment of SEV vulnerability. The spatial assessment of SEV in the areas where it is requested must depend not only on the financial resources for research but also on the availability of opensource software with the functionalities of spatial statistics, such as QGIS, GeoDa or IlWIS. Authors combine traditional and new data sources, spatial variables and indicators, methods, indexes and tools including the temporal dimension, increasing the resolution to the local level with the aim to increase the accuracy and reduce the uncertainty of spatial assessments of SEV related to geohazard

in urban environments.

## 6 Recommendations

The development of a global spatial index of SEV is an urgent task, with the aim of making informed decisions about priority in funding prevention and mitigation actions related to geohazards in urban environments. In the meantime, the priority for these types of assessments must be allocated to developing countries with high population density such as Bangladesh, Haiti,

Philippines, Puerto Rico, el Salvador and Pakistan. More spatial assessment of SEV due to volcanic eruptions and tsunamis in urban environments are needed, but also due to soil erosion and land degradation in the rural zones. Furthermore, the priority must be to allocate funding for countries with high SEV to enable the update of their census information, as this is the most frequent source of secondary data for any SEV assessment. It is also important to encourage the population to share information

through social media (SM) about the vulnerable conditions in which they live, putting in practice the concept of citizen as a sensor (Cervone and Hultquist, 2018).

An assessment of SEV is a condition for the effective development of emergency management capabilities and to reduce the overall time for social recovery after an earthquake (Aubrecht et al., 2013; Garcia et al., 2016). Likewise, spatial assessments of SEV must be considered before taking resettlement decisions for not creating again spatial conditions that favour the SEV. Authors such as Turvey (2007), Walker et al. (2014), Zhou et al. (2014) and Gautam (2017) highlight the need for place-specific, sub-provincial-level, neighbourhood-scale, or local level vulnerability indexes, due to geographic variations in population composition and social structures (Bell N et al., 2007). The macro-scale socio-economic assessment identifies general patterns but fails to capture the detail of the heterogeneity at the micro-scale. Thus, assessment at the provincial, county or state level can result in lost information (Zhou et al., 2014) or requires tackling issues such as ecological fallacy or MAUP (McLaughlin et al., 2002; Openshaw, 1983; Pacione, 2005). In the spatial assessment of SEV, it is necessary to go beyond the administrative boundaries or cartographic variables, with methods such as the dasymetric population mapping (Garcia et al., 2016; Yuan et al., 2019a), square mesh (Renard, 2017), pockets (Lin and Hung, 2016), or *geon* (Kienberger et al., 2009). We found interesting spatial indicators of SEV, such as population density based on land use (Zeng et al., 2012), which we consider more accurate than population density estimated at an area unit. This indicator can better integrate, using RS, the spatial dimension of the exposure and susceptibility of the population in the assessment of the SEV of a case study area. To improve the accuracy and reduce the uncertainty in spatial assessments of SEV must always be the aim. The presence of urban facilities must be included in the assessment of SV. Walker et al. (2014) suggest developing a weighted 'local resource' index for assessing systemic vulnerability since, for example, the absence of sports facilities is associated by Iguacel et al. (2018), Vandermeerschen, Vos, & Scheerder (2015), and Aguilar-Palacio, Gil-Lacruz and Gil-Lacruz (2013) with high levels of SV. In the spatial assessment of SEV, it is also necessary to consider the influence of the spatial component represented by physical space in the degree of vulnerability of a specific area, such as the relationship between slums and a low degree of wellness and health (Buzai and Villerías Alarcón, 2018).

It is necessary to take advantage of the versatility of methods such as ANN based on machine learning to make progress in the spatial assessment of SEV and SMCE in order to map multiple scenarios to inform urban communities and to integrate them in the decision making processes. Communities respond differently to vulnerability maps depending on the purpose behind the maps or the cultural background of the community. On the one hand, some communities reject being mapped as 'victims', but

on the other hand, some request being identified as highly vulnerable to gain access to funding opportunities for activities of risk management (Fekete, 2012). The Walk Score® index developed by Bereitschaft (2017a) although originally orientated to measure only neighbourhood walkability (Bereitschaft, 2017b), can be used a proxy index of spatial SEV in 3D at microscale urban level. The advantage over the SoVI® is that while the SoVI® can be spatialised, Walk Score® is a 3D high resolution spatial index *per se*. The use of the local scale for the assessment of SV will be more useful for the planning of resilient actions

(Lee, 2014; Maharani et al., 2016) than would be vulnerability assessment at a regional scale, which is more orientated to the collection of pathologies in the social dimension. It is necessary to more closely examine so-called 'proxy indicators' to measure spatial SEV at micro-local scales or intra-city levels (Gu et al., 2018). The right management of the spatial component by a community can reduce its economic vulnerability. Groß (2017) presented the case of ski-lift entrepreneurs in Vorarlberg (Austria) who reduced the probability of business interruption by accelerating the uphill and downhill flows of people through

manipulating snow and topography. Regarding tools, it is necessary to take full advantage of the functionalities of opensource software such as and QGIS and ILWIS to make the spatial assessment of SEV to the reach of all the scientific communities around the world.

**Acknowledgements**

The authors thank the National Agency of Research and Development (ANID by its acronym in Spanish) in Chile, which has funded the Research Center for Integrated Disaster Risk Management (CIGIDEN), ANID/FONDAP/15110017 and the Fondecyt Project 1181754/FONDECYT/ANID. Authors also acknowledge the funding received from Engineering and Physical Sciences Research Council (EPSRC) [Grant No. EP/P025951/1] in UK to conclude this research. We want to extend our most thanks to Dr Carolina Martinez for the literature references suggested. We also appreciate the feedback received from

Dr Magdalena Vicuña, Dr Cristina Vizconti, Dr Luis Maldonado and Marta Contreras, BCE. for their feedback during the review process. We also would like to thank the anonymous reviewers and Editors Prof. Dr Thomas Glade and Dr Heidi Kreibich for their contributions to the improvement of this research outcome and Dr Anne Bliss for her support with the English proofreading.

**Funding**

Research Center for Integrated Disaster Risk Management (CIGIDEN), ANID/FONDAP/15110017

Fondecyt Project 1181754/FONDECYT/ANID

Engineering and Physical Sciences Research Council (EPSRC) [Grant No. EP/P025951/1]

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
