# Peer review of "The spatial dimension in the assessment of urban socio-economic vulnerability related to geohazards, a systematic review"

_Natural Hazards and Earth System Sciences, 2019_

## Referee Comment (RC1) · Anonymous Referee #1 · 25 Jul 2019

Using one of the leading databases on scientific journal publications, the authors performed an assessment of articles published between 2008 and 2018 addressing the social and economic dimensions of vulnerability. From originally 235 articles, 21 were finally chosen for a detailed analysis. However, while the authors initially stated to undertake a systematic review for an application on an urban scale, the results and conclusions do not necessarily mirror this aim. From their final choice of contributions, the main conclusion of the authors is that for assessing social vulnerability it is not sufficient to only compute a specific level of vulnerability, but also to include other spatial

information available in order to avoid the modifiable areal unit problem (e.g., Unwin 1996).

Overall, the manuscript has major shortcomings which will be exemplarily addressed in the following list:

- The overall choice of keywords and exclusion of other keywords results in the fact that many studies addressing social vulnerability and/or economic dimensions of vulnerability have not been considered by the authors, which in turn restricts the overall conclusions possible.

- The time period covered is not justified.

- The overall aim to provide a structured overview on studies and indicators, which is not only promised in the title of the contribution but also in the introduction, is not mirrored by the main text body. Materials, methods and findings are rather compiled in a very unstructured way which makes a structured conclusion quite challenging.

- The authors further argue that the economic dimension of vulnerability is the predisposition for the loss of economic value (page 2, lines 15/16), which according to my experience is exactly the contrary relationship – also here we do have scholarly articles which did not make it to the current overview. One reason is again the choice of keywords (see below).

- In the introduction it becomes not clear which specific research question should be answered, and were the niche and the gap for the contribution is to be found. Paragraphs addressing common sense are somehow not connected to those showing specific issues; to give an example it remains unclear why paragraph 3 immediately starts with the SoVI as one of the indices available to assess social vulnerability. On page 3, line 18 the authors even conclude (or state) that only few authors have elaborated on the spatial dimension of social vulnerability, which is wrong if a proper literature research would have been undertaken. There are lots of studies around on this topic, some of

[Figure]

them even in the target journal NHESS. Moreover, the statement that spatial vulnerability assessments only became prominent after the 2004 Indian Ocean Tsunami is neither proven by references, nor true according to my own knowledge. Further, the use of GIS is not only very suitable for assessing spatially the issues of social and economic vulnerability, it is a tool to exactly do this. Finally, the statement that vulnerability is dynamic and subject to spatial and temporal dynamics across scales is not very innovative, there are even specific research papers on this topic from the period 2008-2018.

- Methods: it remains totally open how the amount of 235 papers initially identified was reduced to the final set of 21 contributions. Moreover, searching only for combinations of "social vulnerability" excludes the amount of (valuable) papers around addressing multiple dimensions of vulnerability – and some of these contributions again can be found in NHESS. Further, the authors state in the text that they excluded terms such as "climate change", "health" and "crime analysis", whereas in Figure 1, much more terms have been excluded. BTW: Why has the term "debris" been excluded? Just to give an example, many studies on (social and economic) vulnerability are related to dynamic flooding such as flash floods and debris flows/torrential hazards (even the mentioned EU-funded project MOVE), these are completely ignored by the authors due to their choice of key words.

- In contrast, some of these hazard types are then mentioned in the results section (page 7, second paragraph).

- Instead of showing which contributions used which methods or indicator groups for assessment, the authors could have shown the challenge of indicator interdependencies, one of the main points of criticism for the SoVI. Simply applying the SoVI does not necessarily result in an overview on social and economic dimensions of vulnerability because of the inherent dependencies between indicators.

- In the discussion section the authors have raised some issues that remain questionable, such as the fact that most of the articles related to flood hazard and social vulnerability have been written by geographers because they may be interested in environmental vulnerability.

These issues are just examples underpinning the overall judgement that this contribution is so far not up to international standards. Although the authors have some interesting arguments, I believe that the manuscript needs further improvement to bring it up to an acceptable level before it can be accepted for publication. To summarise, it is not clear why the authors chose specific keywords and excluded others, it is not clear why the authors chose the distinct time period between 2008 and 2018 (the discussion on multiple dimensions of vulnerability and the spatiality of vulnerability is much older). The results are not presented in a logical and organised manner, and the conclusions are not underpinned by the results, some of them seem rather driven by speculation than by evidence.

Therefore, I cannot recommend publication at the current state.

Reference mentioned:

Unwin DJ (1996) GIS, spatial analysis and spatial statistics. Progress in Human Geography 20 (4):540-551. doi:https://doi.org/10.1177/030913259602000408

---

## Referee Comment (RC2) · Anonymous Referee #2 · 24 Dec 2019

Comments This is an important paper to reflect about the current stance of SV research as it is a rare systematic review covering a decade and the period of 2008-2018. The focus is on social and economic aspects of SV indices and their relation to spatial aspects, specifically.

The findings are based upon a systematic selection of studies, yet are also facing certain limitations, obviously, when describing numbers of findings according to countries or even continents, based on a total number of 21 papers only. Shortcomings and guidance for fellow researchers should be added such as the process of selection these papers (why those keywords were selected and others deselected), certain countries and languages that might be overlooked, grey literature and their importance for SV publications such as global or country wise indicator sets, EU is just an example, where many more studies might be found than just within journal papers published on Clarivate Analytics. A discussion section describing those shortcomings would balance out the impression conveyed by this paper that the charts and tables represent world rankings.

Certain older literature might be interesting to add that were dealing with spatial aspects of indicators such as King 2001. The discussion of spatial aspects could also include literature on known effects of spatial indicators per se - for example, within the literature on social-ecological systems that is mostly absent in this selection, but that does deal with socio-economic components of vulnerability. Scale discussions, effects such as the Modifiable Area Unit Problem, ecological fallacy, could be mentioned - even their absence in the literature cited could be of interest. It could also be cautioned more explicitly that while many aspects such as Moron I tests are not mentioned within the journal papers selected does not mean they're not treated by their studies. Often, more technical analyses in GIS such spatial autocorrelation tests are subject to more technical chapters or even shifted into the appendices within project reports or PhD theses.

The authors might connect their review of studies related to the Hyogo Framework also with current strategies such as the Sendai Framework and their related data bases and indices; what aspects of the spatiality of SV are demanded for by those frameworks and which aspects are captured for instance by their indicators or certain other world risk indices? Maybe the findings of the paper could also be compared to findings of similar reviews in terms of predominance of certain factors of vulnerability, prevalence of countries or aspects of spatiality such as scale, unit effects (administrative versus grids, catchments etc), or, shortcomings (de Sherbinin, Fekete, Kuhlicke, Rufat, Tate, Terti are just examples - also look at recent articles). Climate change research

has been excluded, but reasons for this not detailed; more review studies could be found here maybe useful for a discussion section, still such as Ford, Gallina, Preston. Embedding this review into a broader background would better help to clarify the contribution of this paper by covering the period from 2008-2018. The selected decade is fine, since it is recent which is important to add and compare it to previous studies - but this comparison is missing, still. However, these suggestions are optional and the paper does not need to be expanded much on this.

Some minor remarks: I suggest to merge short paragraphs

Section 1 maybe more dimensions of vulnerability should be named and argued, why they had been deselected such as ecological vuln, physical, institutional etc.

Section 2 methods is very short; maybe some more information could be provided such as why a decade has been selected (making it comparable with similar studies such as...?). Clarivate Analytics has been selected, because,....

Section 3: Make consistent use of "I" or "We" How did you define whether the articles were "highly relevant" or "medium"...?

---

## Author Comment (AC1) · 1 Feb 2020

**Reviewer 2**

Thank you very much for your observations. You kindly spent time delving into our manuscript, and we are grateful. Please find your comments in grey, and the respective answers in black. The corresponding paragraph in the paper is in dark blue.

**Comments**

This is an important paper to reflect about the current stance of SV research as it is a rare systematic review covering a decade and the period of 2008-2018. The focus is on social and economic aspects of SV indices and their relation to spatial aspects, specifically.

- The findings are based upon a systematic selection of studies, yet are also facing certain limitations, obviously, when describing numbers of findings according to countries or even continents, based on a total number of 21 papers only.

  Thank you for your comment. As depicted in the methodology, from an initial set of 235 papers, 84 were found to be relevant, 42 of which were considered highly relevant, and 21 were finally reviewed. Nevertheless, we agree that in a revised version of the manuscript, we can consider the complete universe of papers (n = 235) for a more holistic statistical analysis.

- Shortcomings and guidance for fellow researchers should be added such as the process of selection these papers (why those keywords were selected and others deselected), certain countries and languages that might be overlooked, grey literature and their importance for SV publications such as global or country wise indicator sets, EU is just an example, where many more studies might be found than just within journal papers published on Clarivate Analytics. A discussion section describing those shortcomings would balance out the impression conveyed by this paper that the charts and tables represent world rankings.

  Thank you very much for your comment. The choice of keywords and the exclusion of other keywords were our decisions as authors to focus mainly on the spatial dimension in the assessment of socio-economic vulnerability related to internal geo-dynamic processes such as earthquakes, tsunamis and volcanic eruptions. We will clarify this aspect in a revised version of the manuscript. We wanted to focus mainly on journal papers rather than other types of literature; however, we are open to checking other databases such as Scopus, in addition to Clarivate. We will also improve the discussion section based on the suggestion from the reviewer. Finally, the references included in the tables represent, according to the judgement of the authors, the most relevant ones regarding data sources, methods, spatial variables, indicators, indexes and tools used for spatial socio-economic vulnerability assessments.

- Certain older literature might be interesting to add that were dealing with spatial aspects of indicators such as King 2001.

  Thank you for this suggestion. We are open to reviewing this reference suggested by you regardless of the publication period.

- The discussion of spatial aspects could also include literature on known effects of spatial indicators per se - for example, within the literature on social-ecological systems that is mostly absent in this selection, but that does deal with socio-economic components of vulnerability. Scale discussions,

effects such as the Modifiable Area Unit Problem, ecological fallacy, could be mentioned even their absence in the literature cited could be of interest.

Thank you for this suggestion. Effects such as the modifiable areal unit problem (MAUP) and ecological fallacy are already discussed in our manuscript. Please read below:

'(…) Thus, assessment at the provincial, county or state level can result in lost information (Zhou, Li, Wu, Wu, & Shi, 2014) or require tackling issues such as ecological fallacy or the modifiable areal unit problem (MAUP) (Pacione, 2005)(…)'. Lines 4-6, page 17.

- It could also be cautioned more explicitly that while many aspects such as Moron I tests are not mentioned within the journal papers selected does not mean they're not treated by their studies. Often, more technical analyses in GIS such spatial autocorrelation tests are subject to more technical chapters or even shifted into the appendices within project reports or PhD theses.

Thank you for your suggestion. We are open to reviewing the references that you suggested. We did not manage to find any references to the Moron I test; however, the manuscript already includes several references to global Moran's I as a statistic method to determine spatial autocorrelation and for the assessment of social vulnerability (SV). Please read below:

'(…) hence, Gu et al., (2018) used global Moran's I and local Gi* de Getis-Ord in addition to the SoVI®, while Maharani, Lee, and Ki, (2016) utilised the SOM (…)'. Line 5-6, pag. 11

'(…) Buzai & Villerías Alarcón (2018) developed their own SV index and also used global Moran's I, but they elaborated on the spatial patterns of local association using the Local Index of Spatial Association (LISA) to determine hot and cold spots (…)'. Line 7-8, pag. 11

'(…) Lin and Hung (2016) combined Gi* de Getis-Ord to measure the high or low vulnerability association and global Moran's I to determine the homogeneity of the clusters (…)'. Line 8-10, pag. 11

'(…) According to Ley-García, Denegri de Dios, & Ortega Villa, (2015) global Moran's I and LISA allow the identification of dependence between attributes and localisations (…)'. Line 10-12, pag. 11

'(…) The summary measure of autocorrelation in the territory as a whole is undertaken with global Moran's I, while the autocorrelation of the spatial units included in the territory is measured using LISA (…)'. Line 12-14, pag. 11

'(…) Cutter and Finch (2008) also previously utilised global Moran's I and LISA to identify local variability and cluster similarity of low and social vulnerability (…)'. Line 14-15, pag. 11

'(…) These authors used global and local Moran's I or LISA as ESDA to determine the spatial autocorrelation amongst counties and identify the similarity and/or dissimilarity in the clustering of SV (…)'. Line 17-18, pag. 11

- The authors might connect their review of studies related to the Hyogo Framework also with current strategies such as the Sendai Framework and their related data bases and indices; what aspects of the spatiality of SV are demanded for by those frameworks and which aspects are captured for instance by their indicators or certain other world risk indices? Maybe the findings of the paper could also be compared to findings of similar reviews in terms of predominance of certain factors of vulnerability, prevalence of countries or aspects of spatiality such as scale, unit effects

(administrative versus grids, catchments etc), or, shortcomings (de Sherbinin, Fekete, Kuhlicke, Rufat, Tate, Terti are just examples - also look at recent articles).

Thank you very much for your suggestion. We decided as authors to focus our systematic review primarily on peer-reviewed papers that address the spatial dimension in the assessment of socio-economic vulnerability related to internal geo-dynamic processes such as earthquakes, tsunamis and volcanic eruptions. The Hyogo and Sendai frameworks are documents prepared to tackle more general aspects of disaster risk reduction (DRR) and the achievement of the sustainable development goals (SDGs), respectively. However, we can refer to both frameworks in the discussion section to highlight the contribution of the manuscript to the implementation of these frameworks. A reference to Fekete included one of the papers considered relevant for the review.

Fekete, A. (2012). Spatial disaster vulnerability and risk assessments: challenges in their quality and acceptance. Natural Hazards, 61(3), 1161-1178. doi:10.1007/s11069-011-9973-7

- Climate change research has been excluded, but reasons for this not detailed; more review studies could be found here maybe useful for a discussion section, still such as Ford, Gallina, Preston.

Thanks for your observation and suggestions. The choice of keywords, as well as the exclusion of other keywords, was our decision as authors to focus mainly on the spatial dimension in the assessment of socio-economic vulnerability related to internal geo-dynamic processes such as earthquakes, tsunamis and volcanic eruptions. We will clarify this aspect in a revised version of the manuscript. The reason 'climate change' was not considered as a search term is that, in Chile, this topic is mainly addressed by the Centre for Climate and Resilience Research (CR)2, and we did not want to step into their research field.

- Embedding this review into a broader background would better help to clarify the contribution of this paper by covering the period from 2008-2018. The selected decade is fine, since it is recent which is important to add and compare it to previous studies - but this comparison is missing, still. However, these suggestions are optional and the paper does not need to be expanded much on this.

Thank you for this suggestion. The reason for selecting the period 2008–2018 was to investigate the state of the art on the topic of the spatial dimension in the assessment of socio-economic vulnerability related to internal geodynamic processes, which we believe has been covered in the past 10 years. We will include this clarification in a revised version of the manuscript.

**Some minor remarks:**

- I suggest merging short paragraphs

Thank you for this suggestion. It will be taken into account in the revised version.

- Section 1 maybe more dimensions of vulnerability should be named and argued, why they had been deselected such as ecological vuln, physical, institutional etc.

Thank you for reminding us that, apart from the social and economic dimensions, the following are also dimensions of vulnerability: physical, cultural, environmental and institutional (Birkmann et al., 2013). Nevertheless, we prefer to focus on the socio-economic dimensions for this specific research.

- Section 2 methods is very short; maybe some more information could be provided such as why a decade has been selected (making it comparable with similar studies such as...?). Clarivate Analytics has been selected, because,....

Thank you for your observation and suggestion. As we explained before, the reason for selecting the period 2008–2018 was to explore the state of the art on the topic of the spatial dimension in the assessment of socio-economic vulnerability related to internal geodynamic processes, which we believe has been covered in the past 10 years. We will include this clarification in a revised version of the manuscript. We selected Clarivate Analytics as the database for undertaking the literature review search because we consider it to be the most complete leading database of scientific journal publications.

- Section 3: Make consistent use of "I" or "We"
  Thanks for your observation. The correct pronoun is 'we'.

- How did you define whether the articles were "highly relevant" or "medium"...?

The main criterion for defining the relevance of a manuscript is the use of spatial variables, indicators and/or indexes for the assessment of socio-economic vulnerability. The initial number of papers selected through the systematic review was reduced based on their relevance to the topic of the spatial assessment of socio-economic vulnerability related to mainly internal geodynamic processes. However, in the final set of papers, we also included those related to hydrometeorological hazards, epidemics and anthropogenic hazards that contain spatial variables, indicators or indexes that could be applied to the spatial assessment of socio-economic vulnerability related to internal geodynamic processes.

**References**

Birkmann, J., Cardona, O. D., Carreño, M. L., Barbat, A. H., Pelling, M., Schneiderbauer, S., . . . Welle, T. (2013). Framing vulnerability, risk and societal responses: the MOVE framework. *Natural Hazards, 67*(2), 193-211. doi:10.1007/s11069-013-0558-5

Buzai, G., & Villerías Alarcón, I. (2018). Análisis espacial cuantitativo de los determinantes sociales de la salud (DSS) en la cuenca del río Luján (provincia de Buenos Aires, Argentina). *Estudios Socioterritoriales, 23*.

Cutter, S. L., & Finch, C. (2008). Temporal and spatial changes in social vulnerability to natural hazards. *Proceedings of the National Academy of Sciences, 105*(7), 2301-2306. doi:10.1073/pnas.0710375105

Gu, H., Du, S., Liao, B., Wen, J., Wang, C., Chen, R., & Chen, B. (2018). A hierarchical pattern of urban social vulnerability in Shanghai, China and its implications for risk management. *Sustainable Cities and Society, 41*, 170-179. doi:https://doi.org/10.1016/j.scs.2018.05.047

Ley-García, J., Denegri de Dios, F. M., & Ortega Villa, L. M. (2015). Spatial dimension of urban hazardscape perception: The case of Mexicali, Mexico. *International Journal of Disaster Risk Reduction, 14*, 487-495. doi:https://doi.org/10.1016/j.ijdrr.2015.09.012

Lin, W.-Y., & Hung, C.-T. (2016). Applying spatial clustering analysis to a township-level social vulnerability assessment in Taiwan. *Geomatics, Natural Hazards and Risk, 7*(5), 1659-1676. doi:10.1080/19475705.2015.1084542

Maharani, Y. N., Lee, S., & Ki, S. J. (2016). Social vulnerability at a local level around the Merapi volcano. *International Journal of Disaster Risk Reduction, 20*, 63-77. doi:https://doi.org/10.1016/j.ijdrr.2016.10.012

Pacione, M. (2005). *Urban geography : a global perspective* (Second edition ed.). London etc.: Routledge.

Zhou, Y., Li, N., Wu, W., Wu, J., & Shi, P. (2014). Local Spatial and Temporal Factors Influencing Population and Societal Vulnerability to Natural Disasters. *Risk Analysis, 34*(4), 614-639. doi:doi:10.1111/risa.12193

---

## Author Comment (AC2) · 1 Feb 2020

**Reviewer 1**

Thank you very much for your observations. You kindly spent time delving into our manuscript, and we are grateful. We have used a colour code to answer your questions. Please find your comments in grey, and the respective answers in black. The corresponding paragraph in the paper is in dark blue.

**Comments**

- Using one of the leading databases on scientific journal publications, the authors performed an assessment of articles published between 2008 and 2018 addressing the social and economic dimensions of vulnerability. From originally 235 articles, 21 were finally chosen for a detailed analysis. However, while the authors initially stated to undertake a systematic review for an application on an urban scale, the results and conclusions do not necessarily mirror this aim.

Thank you for your observation. A systematic review searches for, appraises and synthesises research evidence (Grant & Booth, 2009). We indicate the time period selected for the systematic review (2008-2018) as well as an indication of the terms selected for the query. As a decision of one of the authors, the complete list of search terms was not included in the manuscript; however, we can include it in a revised version. Please find the search terms used in Table 1.

| Topic | (social vulnerability* OR societal vulnerability* OR socioeconomic vulnerability* OR socio-economic vulnerability* OR economic vulnerability*) |
|---|---|
| | AND |
| Topic | (area* OR distance* OR range* OR distance* OR direction* OR spatial geometries* OR patterns* OR spatial connectivity* OR isolation* OR diffusion* OR spatial association* OR scale* OR accessibility* OR network* OR cluster*) |
| | NOT |
| Topic | (climate change* OR ecological* OR drought* OR resilience* OR debris* OR epidemiological* OR substance* OR behavioral* OR evacuation* OR recovery* OR pollution* OR leptospirosis* OR violence* OR illness* OR disease* OR heat* OR crisis* OR Conflict* OR deaths* OR obesity* OR criminal* OR chemical* OR symptoms* OR syndrome* OR food insecurity* OR air pollution* OR stress* OR diabetes* OR depressive* OR alcohol* OR cancer* OR drugs* OR palm oil* OR tobacco* OR smoke* OR storm* OR psychometric* OR cocaine* OR toxic* OR palliative* OR therapy* OR HIV* OR dengue* OR ecosystem* OR rheumatoid arthritis* OR nutritional* OR malaria* OR resources* OR sexual activity* OR sexual health*). |

Table 1. Search terms used in the systematic review.

- From their final choice of contributions, the main conclusion of the authors is that for assessing social vulnerability it is not sufficient to only compute a specific level of vulnerability, but also to include other spatial information available in order to avoid the modifiable areal unit problem (e.g., Unwin 1996).

Thanks for your comment. The main conclusion of our manuscript is as follows:

'(…) we can conclude that it is not sufficient to only estimate the specific level of vulnerability per unit area; it is also necessary to determine the influence of the spatial component in this degree of socio-economic vulnerability(…)'.

Rather than the modifiable areal unit problem (MAUP), we wanted to make reference to 1) the influence of the elements and their configuration on a physical space that contributes to reducing or decreasing the degree of vulnerability of a specific area, such as the relationship between slums

and a low degree of wellness and health (Buzai & Villerías Alarcón, 2018)', 2) the Walk Score® index (Bereitschaft, 2017a) walkability (Bereitschaft, 2017b) and 3) the manipulation of snow and topography in Vorarlberg, Austria (Groß, 2017). With this approach, we wish to go beyond theory and demonstrate a more tangible relationship between socio-economic vulnerability and space in an urban environment.

- The overall choice of keywords and exclusion of other keywords results in the fact that many studies addressing social vulnerability and/or economic dimensions of vulnerability have not been considered by the authors, which in turn restricts the overall conclusions possible.

Thanks for your observation. The choice of keywords, as well as the exclusion of other keywords, was our decision as authors to focus mainly on the spatial dimension in the assessment of socio-economic vulnerability related to internal geo-dynamic processes such as earthquakes, tsunamis and volcanic eruptions. We will clarify this aspect in a revised version of the manuscript. The reason we did not consider climate change as a search term is that this topic is mainly addressed by the Centre for Climate and Resilience Research (CR)2 in Chile, and we did not want to step into its research field.

- The time period covered is not justified

Thank you for this observation. The reason for selecting the period 2008–2018 was to explore the state of the art on the topic of the spatial dimension in the assessment of socio-economic vulnerability related to internal geodynamic processes, which we believe has been covered in the past 10 years. We will include this clarification in a revised version of the manuscript. Of course, we are open to reviewing other references suggested by you, regardless of the publication period. Thank you for your suggestion.

- The overall aim to provide a structured overview on studies and indicators, which is not only promised in the title of the contribution but also in the introduction, is not mirrored by the main text body. Materials, methods and findings are rather compiled in a very unstructured way which makes a structured conclusion quite challenging.

Thank you for your observation. We are afraid that we do not have a materials section. The methods, as well as the data sources, spatial variables, indicators, indexes and tools, which we believe you named 'findings', are listed in the following tables:

Table 2. Data sources for the spatial assessment of socio-economic vulnerability.
Table 3. Methods applied to the spatial assessment of socio-economic vulnerability.
Table 4. Spatial variables for socio-economic vulnerability assessments.
Table 5. Spatial indicators for socio-economic vulnerability assessments.
Table 6. Spatial indexes for socio-economic vulnerability assessments.
Table 7. Tools used for spatial socio-economic vulnerability assessments.

Tables 2, 3, 4, 5 and 6 are structured mainly in two columns: the first column lists data sources, methods, spatial variables, spatial indicators, and spatial indexes, respectively; the second column contains the authors and the year of their publications, in which the mentioned topics are addressed. Moreover, the references are listed from the most recent publication to the oldest ones in the period from 2008 to 2018. Table 7 about tools includes three columns, namely, method, software and authors, and the structure and the time period covered are the same as the previous tables. Please find below each of the tables mentioned:

[revised manuscript text omitted]

- The authors further argue that the economic dimension of vulnerability is the predisposition for the loss of economic value (page 2, lines 15/16), which according to my experience is exactly the contrary relationship – also here we do have scholarly articles which did not make it to the current overview. One reason is again the choice of keywords (see below).

Thanks for this comment. We respect your opinion; however, based on the previous work of one of the authors, we prefer to stick with the definition of economic dimension of vulnerability formulated by Birkmann et al. (2013): 'Economic dimension: propensity for loss of economic value from damage to physical assets and/or disruption of productive capacity'. We consider that the opposite concept will be more related to economic resilience than vulnerability.

- In the introduction it becomes not clear which specific research question should be answered and were the niche and the gap for the contribution is to be found.

Thank you for your observation and question. The research question was,'Which spatial variables/indicators/indexes are useful to characterise the socio-economic vulnerability to natural

hazards in urban environments?' In a revised version of the manuscript, we will include a rephrased version of the research question as follows:'Which spatial variables/indicators/indexes are useful to characterise the socio-economic vulnerability to internal geodynamic processes in urban environments?'

▪ Paragraphs addressing common sense are somehow not connected to those showing specific issues; to give an example it remains unclear why paragraph 3 immediately starts with the SoVI as one of the indices available to assess social vulnerability.

Thanks for this observation. After checking carefully, we found the first reference to SoVI® in the manuscript in line 20 on the second page. The reason for including this reference to the index for the assessment of social vulnerability (SV), developed by Cutter, Boruff and Shirley (2003), is that in previous lines (17 to 19), we described the object of the assessment of SV: (…) 'The assessment of SV is orientated to cast the light on the most susceptible groups of a population to be impacted by a disaster, in the spatial and temporal dimensions (Zhou et al., 2014)'. Then, we decided that the next paragraph should start with the first index developed to assess SV, which, to our best knowledge, is the SoVI®.

▪ On page 3, line 18 the authors even conclude (or state) that only a few authors have elaborated on the spatial dimension of social vulnerability, which is wrong if proper literature research would have been undertaken. There are lots of studies around on this topic, some of them in the target journal NHESS.

Thanks for your opinion. We do agree that there are several studies related to the spatial dimension of socio-economic vulnerability, but not specifically related to internal geodynamic processes. Regarding references from the NHESS journal, we are happy to highlight that one of the references from the present journal is already considered in the manuscript (Lines 21 -21, page 11): '(…) The *geon* approach also identifies clusters using semi-automated regionalisation in multispectral image data to represent a socioeconomic vulnerability in the form of spatial vulnerability units (SVU) (Kienberger, Lang, & Zeil, 2009) (…)'.

▪ On page 3, line 18 the authors even conclude (or state) that only few authors have elaborated on the spatial dimension of social vulnerability, which is wrong if a proper literature research would have been undertaken. There are lots of studies around on this topic, some of them even in the target journal NHESS.

Thank you for your observation. We offer to rephrase the following statement: '(…) only a few authors in the period between 2008 to 2018 have elaborated on the spatial dimension of socio-economic vulnerability related to internal geodynamic processes (…)'. This could be a conclusion

as well as the basis for a recommendation to conduct further research into the mentioned aspect. In addition, we are willing to redo the systematic search of relevant literature references that we could have missed, including but not limited to the NHESS journal.

- Moreover, the statement that spatial vulnerability assessments only became prominent after the 2004 Indian Ocean Tsunami is neither proven by references, nor true according to my own knowledge.

Thank you for this observation. We acknowledge that we drafted this sentence incorrectly. Citing Fekete (2012), we wanted to state that events such as the Indian Ocean tsunami in 2004 and Hurricane Katrina in 2005, with an explicit spatial component, sparked again the research community's interest in those social groups that are more affected by this type of phenomena. We will rephrase the sentence accordingly:

'(…) the Indian Ocean tsunami in 2004, as a result of its large impact area, sparked again the research community's interest in spatial vulnerability analyses and an interdisciplinary approach, which illuminated the problems faced by low-income populations after disasters (…)'.

- Further, the use of GIS is not only very suitable for assessing spatially the issues of social and economic vulnerability, but it is also a tool to exactly do this.

Thank you for this observation. While we agree with your claim, after carefully going through the manuscript, we found that the exact statement in the manuscript is (line 4, page 4): '(…) The use of geographic information systems (GIS) to collect and process data related to hazards and vulnerability was found very suitable (Fekete, 2012) (…)'. The reason for including this statement is that, in the past, hazard and vulnerability data collection processes were performed manually, making the assessment highly time-consuming. Now, these hazard and vulnerability assessments are speeded up with the integration of GIS into the process. However, the potential of GIS is sometimes untapped and limited to the mapping of the socio-economic characteristics of a case study area, without taking into account the influence of the spatial component that can be integrated to take advantage of the GIS capabilities.

- Finally, the statement that vulnerability is dynamic and subject to spatial and temporal dynamics across scales is not very innovative, there are even specific research papers on this topic from the period 2008-2018.

Thank you for this comment. We agree with the reviewer. Nevertheless, the objective of this manuscript is to determine the spatial variables, indicators and indexes used to characterise socio-economic vulnerability to internal geodynamic processes in the period between 2008 and 2018 in urban environments; the manuscript will be a guide for scientists who wish to perform a spatial assessment.

- Methods: it remains totally open how the amount of 235 papers initially identified was reduced to the final set of 21 contributions.

Thanks for your observation. The initial number of papers selected through the systematic review was reduced based on the relevance to the topic of spatial assessment of socio-economic vulnerability related to mainly internal geodynamic processes. However, in the final set of papers, we also included those related to hydrometeorological hazards, epidemics and anthropogenic hazards that contain spatial (Gu et al., 2018) variables, indicators or indexes that could be applied to the spatial assessment of socio-economic vulnerability related to internal geodynamic

processes. Therefore, following your observation, we will include this explanation in the manuscript.

▪ Moreover, searching only for combinations of "social vulnerability" excludes the amount of (valuable) papers around addressing multiple dimensions of vulnerability – and some of these contributions again can be found in NHESS.

Thank you for reminding us that, apart from the social and economic dimensions, other dimensions of vulnerability also exist physical, cultural, environmental and institutional (Birkmann et al., 2013). Nevertheless, we prefer to focus on the social and economic dimensions for this specific research.

▪ Further, the authors state in the text that they excluded terms such as "climate change", "health" and "crime analysis", whereas in Figure 1, much more terms have been excluded. BTW: Why has the term "debris" been excluded? Just to give an example, many studies on (social and economic) vulnerability are related to dynamic flooding such as flash floods and debris flows/torrential hazards (even the mentioned EU-funded project MOVE), these are completely ignored by the authors due to their choice of key words.

Thank you for your suggestion; however, we are afraid that the term 'debris' suggested by you is not a spatial variable, indicator or index that is useful for the assessment of socio-economic vulnerability. The area where the debris appears distributed after a flash flood will be more useful for damage and exposure estimation and/or hazard zonation because of floods than for a complete socio-economic vulnerability assessment related to internal geodynamic processes.

▪ In contrast, some of these hazard types are then mentioned in the results section (page 7, second paragraph).

Thanks for your observation. In this manuscript, we already stated that we are focused on the topic of the spatial assessment of socio-economic vulnerability related to mainly internal geodynamic processes; however, in the final set of selected papers, we also included those related to hydrometeorological hazards, epidemics and anthropogenic hazards that contain variables, indicators or indexes that could be applied to the spatial assessment of socio-economic vulnerability related to internal geodynamic processes.

▪ Instead of showing which contributions used which methods or indicator groups for assessment, the authors could have shown the challenge of indicator interdependencies, one of the main points of criticism for the SoVI. Simply applying the SoVI does not necessarily result in an overview on social and economic dimensions of vulnerability because of the inherent dependencies between indicators.

Thank you very much for this comment. We agree with you. Therefore, in addition to the information that is currently in the manuscript, we will include your comment as a conclusion. Moreover, as a recommendation, we will suggest using the stepwise regression analysis to avoid collinearity between variables and/or indicators, removing the weakest correlated variables and spatial indicators and identifying those that best explain the socio-economic vulnerability of a particular area to take actions to reduce it.

- In the discussion section the authors have raised some issues that remain questionable, such as the fact that most of the articles related to flood hazard and social vulnerability have been written by geographers because they may be interested in environmental vulnerability.

Thanks for this observation. It was a statement based on the evidence collected during the systematic review, but as we are mainly focused on the spatial assessment of socio-economic vulnerability related to mainly internal geodynamic, we can delete that sentence.

- These issues are just examples underpinning the overall judgement that this contribution is so far not up to international standards. Although the authors have some interesting arguments, I believe that the manuscript needs further improvement to bring it up to an acceptable level before it can be accepted for publication.

Thanks for the assessment of our manuscript. We expect that based on your comments, we will be able to produce a revised version that meets international standards and can, therefore, be published.

- To summarise, it is not clear why the authors chose specific keywords and excluded others, it is not clear why the authors chose the distinct time period between 2008 and 2018 (the discussion on multiple dimensions of vulnerability and the spatiality of vulnerability is much older). The results are not presented in a logical and organised manner, and the conclusions are not underpinned by the results, some of them seem rather driven by speculation than by evidence.

The criteria to select the search terms were those spatial variables, indicators and indexes useful for assessing socioeconomic vulnerability mainly related to internal geodynamic processes. The reason for selecting the period 2008–2018 was to examine the state of the art on the topic of the spatial dimension in the assessment of socio-economic vulnerability related to internal geodynamic processes, which we considered to have been covered in the past 10 years. The results regarding more frequent methods, as well as the data sources, spatial variables, indicators, indexes and tools used for the assessment of socio-economic vulnerability related to internal geodynamic processes, are listed in tables. These findings support most of the conclusions.

---

## Author Response (AR1)

Dear Editor,

Thank you very much for your summary of the reviewers' comments. We are grateful for your thorough assessment of our manuscript. To respond to your comments, we have used a colour code: your comments and questions already solved are in grey and our respective answers in black. The corresponding paragraph in the paper is dark blue.

**Comments**

- As you know, two reviewers have now provided detailed reviews, which you have replied in detail to. Despite the fact that, that one reviewer recommended to reject your manuscript, I offer the possibility to revise the manuscript. I find your topic very interesting and I am optimistic, that you will be able to significantly and sufficiently improve your manuscript with major revision. However, I expect that quite some effort is necessary.

  We appreciate the opportunity to revise our manuscript. We are confident that we have addressed this important topic by following the comments from you and your reviewers The revision required quite a lot of effort, but the result is a more consistent paper.

- One major critique is, that both reviewer doubt that you identified all relevant papers for your research question and focus of review. Of course, the selection of key words for the search are your decision, however, it is also your responsibility to make sure that you identify all relevant papers with your selection of key words.

  Thanks for this observation. Following this comment and the comments of the reviewers, we decided to re-run the search query, updating the time period for the systematic review from 2008-2018 to 2010 – 2020. The reason to update the time period is that in this systematic review we are interested in knowing the state of the art in data sources, spatial variables, indicators, methods, indexes and tools for the assessment of the socio-economic vulnerability (SEV) related to geohazards, which we consider is covered in the last ten years.

  To ensure that we identified all relevant papers for our research question, this time we undertook the search not only in Clarivate but also in two more databases: Scopus/Elsevier and Google Scholar. Additionally, to the different names for socio-economic vulnerability (SEV), and spatial variables, we added the complete list of geohazards that we are interested in this research to the query: "earthquakes" OR "tsunamis" OR "volcanic eruptions" OR "landslides" OR "soil erosion" OR "land degradation".  Following the request of the reviewers, we included in this revised version the table with the terms included and excluded to select relevant literature reference, that was not included in the previous version as a suggestion of the second co-author. Please find the search terms considered in Table 1.

| D | Q | SEARCH TERMS |
|---|---|---|
| Clarivate analytics | TOPIC | "social vulnerability" OR "economic vulnerability" OR "socioeconomic vulnerability" OR "socio-economic vulnerability" |
| | | AND |

| D | Q | SEARCH TERMS |
|---|---|---|
| | TOPIC | "area" OR "distance" OR "range" OR "distance" OR "direction" OR "spatial geometries" OR "patterns" OR "spatial connectivity" OR "isolation" OR "diffusion" OR "spatial association" OR "scale" OR "accessibility" OR "network" OR "cluster" |
| | | AND |
| | TOPIC | "earthquakes" OR "tsunamis" OR "volcanic eruptions" OR "landslides" OR "soil erosion" OR "land degradation" |
| | | NOT |
| | TOPIC | "climate change" OR "ecological" OR "drought" OR "resilience" OR "debris" OR "epidemiological" OR "substance" OR "behavioural" OR "evacuation" OR "recovery" OR "pollution" OR "leptospirosis" OR "violence" OR "illness" OR "disease" OR "heat" OR "crisis" OR "conflict" OR "deaths" OR "obesity" OR "criminal" OR "chemical" OR "symptoms" OR "syndrome" OR "food insecurity" OR "air pollution" OR "stress" OR "diabetes" OR "depressive" OR "alcohol" OR "cancer" OR "drugs" OR "palm oil" OR "tobacco" OR "smoke" OR "storm" OR "psychometric" OR "cocaine" OR "toxic" OR "palliative" OR "therapy" OR "HIV" OR "dengue" OR "ecosystem" OR "rheumatoid" "arthritis" OR "nutritional" OR "malaria" OR "resources" OR "sexual activity" OR "sexual health" |
| Scopus/Elsevier | Article title, abstract, keywords | (TITLE-ABS-KEY ( "social vulnerability*" AND "economic vulnerability*") AND TITLE-ABS-KEY ("socioeconomic vulnerability*") AND TITLE-ABS-KEY ("area" OR "distance" OR "range" OR "distance" OR "direction" OR "spatial geometries" OR "patterns" OR "spatial connectivity" OR "isolation" OR "diffusion" OR "spatial association" OR "scale" OR "accessibility" OR "network" OR "cluster") AND TITLE-ABS-KEY ("earthquakes" OR "tsunamis" OR "volcanic eruptions" OR "landslides" OR "soil erosion" OR "land degradation") AND NOT TITLE-ABS-KEY ("climate change" OR "ecological" OR "drought" OR "resilience" OR "debris" OR "epidemiological" OR "substance" OR "behavioral" OR "evacuation" OR "recovery" OR "pollution" OR "leptospirosis" OR "violence" OR "illness" OR "disease")) AND DOCTYPE (ar) AND PUBYEAR > 2009 AND PUBYEAR <2021 |

D: Database
Q: Query

Table 1. Terms included and excluded to select relevant literature references in Clarivate analytics.

Google scholar was discarded as a source of references given that when the query was run on this database, it provided only one reference, which was the first version of this manuscript. Please see the evidence below in Figure 1.

[Figure]

Figure 1. Evidence of the result of the query on the Google Scholar database.

- Since you want to provide a stat of the art review, I don't think that it is a good idea to exclude all most recent papers, i.e. the ones published after 2018. Additionally, I suggest that you also present results of key papers, which were published before 2008, maybe in a sort of background or in the introduction chapter.

Thank you for your suggestion. We could not agree more with this observation. In consequence, we decided to re-run the search query, updating the time period for the systematic review from 2008-2018 to 2010 – 2020. This time the search query allowed us to identify five references published after 2018:

- o Aksha, S. K., Resler, L. M., Juran, L., & Carstensen, L. W. (2020). A geospatial analysis of multi-hazard risk in Dharan, Nepal. Geomatics Natural Hazards & Risk, 11(1), 88-111. doi:10.1080/19475705.2019.1710580
- o Kurnianto, F. A., Ikhsan, F. A., Apriyanto, B., & Nurdin, E. A. (2019). Earthquake vulnerability disaster in the Lembang district of West Bandung Regency, Indonesia. Earthquake Science, 32(1), 40-46. doi:10.29382/eqs-2019-0040-5
- o Muir, J. A., Cope, M. R., Angeningsih, L. R., Jackson, J. E., & Brown, R. B. (2019). Migration and Mental Health in the Aftermath of Disaster: Evidence from Mt. Merapi, Indonesia. International Journal of Environmental Research and Public Health, 16(15), 19. doi:10.3390/ijerph16152726
- o Rezaei-Malek, M., Torabi, S. A., & Tavakkoli-Moghaddam, R. (2019). Prioritizing disaster-prone areas for large-scale earthquakes' preparedness: Methodology and application. Socio-Economic Planning Sciences, 67, 9-25. doi:10.1016/j.seps.2018.08.002
- o Yuan, H. H., Gao, X. L., & Qi, W. (2019). Fine-Scale Spatiotemporal Analysis of Population Vulnerability to Earthquake Disasters: Theoretical Models and Application to Cities. Sustainability, 11(7), 19. doi:10.3390/su11072149

This time the oldest references are from 2012 and they are only two, one identified with the current refined search query and another identified in the previous search. Following your suggestion, we also reviewed relevant references published before 2008 such as:

[revised manuscript text omitted]

These references were incorporated into the manuscript, not only in the introduction but also in the discussion of results. Please see the texts as follows:

[revised manuscript text omitted]

- Please improve the definition of your research question, and on this basis the criteria on how you reduced the >200 papers to 21. This is a critical point in your method, so this selection needs to be very well justified and transparent.

Thanks for this observation. Following your request, we rephrased our research question to: what is the state of the art in the spatial assessment of SEV to geohazards in urban environments? Based on the refined research question, we refined also our search query including new key words, which were already listed in Table 1.

The gross number of articles identified using the search query were 29, having two matching references in Clarivate analytics and Scopus/Elsevier: Kurnianto et al., (2019) and Eidsvig (2014). Thus, eventually, we identified 27 references. Despite the precise search query, 11 references were discarded. The detailed reasons to rule these references out is explained in Table 2, but this is only for the information of the editor, a summarized version of the reasons explained in Table 2 will be in the final version of the manuscript.

| | REFERENCE | YEAR | REASONS TO BE DISCARDED |
|---|---|---|---|
| 1 | Papathoma-Kohle, M., Cristofari, G., Wenk, M., & Fuchs, S. | 2019 | It presents a couple of indexes (PTVA-3 and PTVA-4) made up of variables in the physical dimension, rather than socio-economic, for the assessment of the vulnerability of buildings to tsunamis in Apulia (Italy). |
| 2 | Yuan, H. H., Gao, X. L., & Qi, W. | 2019 | Another reference from the same authors was already selected for the review: Yuan, H. H., Gao, X. L., & Qi, W. |

| | REFERENCE | YEAR | REASONS TO BE DISCARDED |
|---|---|---|---|
| | | | (2019). Fine-Scale Spatiotemporal Analysis of Population Vulnerability to Earthquake Disasters: Theoretical Models and Application to Cities. Sustainability, 11(7), 19. doi:10.3390/su11072149 |
| 3 | Zhang, N., & Huang, H. | 2018 | This paper provides a Gaussian blur-based method to calculate the average severity of disasters, rather than focusing on methods for the spatial assessment of SEV. |
| 4 | Shen, S., Cheng, C. X., Yang, J., & Yang, S. L. | 2018 | This reference analyses the trends and hot topics in disaster research in recent years. 'social vulnerability' is identified as one of the hot topics, but the authors do not go deep into this topic in the paper. |
| 5 | Goncalves, M., & Vizintim, M. F. B. | 2017 | This article is written in Portuguese and none of the authors are proficient in this language. |
| 6 | Karagiorgos, K., Heiser, M., Thaler, T., Hubl, J., & Fuchs, S. | 2016 | This paper addresses the topic of flash floods, which is a hazard not included in the search query. |
| 7 | Postiglione, I., Masi, A., Mucciarelli, M., Lizza, C., Camassi, R., Bernabei, V., . . . Peruzza, L. | 2016 | This reference elaborates on an Italian communication campaign oriented to prevent or at least reduce the risk associated with earthquakes with the aim to promote a culture of seismic risk prevention starting with volunteers to serve later as multipliers. Prevention campaigns due to seismic hazard are out of the scope of our manuscript. |
| 8 | Alcántara-Ayala, I., & Oliver-Smith, A. | 2014 | This article presents the activities undertaken by the ICL Latin-American network (ICLLAN) related to capacity building to reduce risk due to landslides through forensic investigations. It deals with one of the geohazards considered in this paper, but not the spatial assessment of SEV. |
| 9 | Khazai, B., Daniell, J. E., Düzgün, Ş., Kunz-Plapp, T., & Wenzel, F. | 2014 | This reference was discarded for two reasons. First, it is focused on modelling shelter needs and health impacts caused by earthquakes rather than the spatial assessment of SEV. Second, it is a book chapter, while the systematic review focuses on articles. |
| 10 | Vilches, O. R., Carrillo, K. S., Reyes, C. M., & Castillo, E. J. | 2014 | This paper aims to evaluate the socio-environmental effects of the 27/10/2010 tsunami in Chile considering the SEV, the safety perception, and the environmental problems, in highly vulnerable rural towns, that depend on the extraction of resources from the sea. Although this is a very interesting topic is neither in the scope of the review nor does it make use of any spatial variable, indicator, or index that could be useful for the present review. |

| | REFERENCE | YEAR | REASONS TO BE DISCARDED |
|---|---|---|---|
| 11 | Jaque Castillo, E., Contreras, A., Ríos, R., & Quezada Flory, J. | 2013 | Although this reference assesses the socio-economic. educational and physical vulnerability due to tsunami in the Town of Tirua (Chile), it does not make use of any spatial, variable, indicator or index useful for the current research. |

Table 2. References discarded for the systematic review.

The summarized version of the reasons to discard the 11 references, although they were identified through the search query, is described in the manuscript as follows:

'(…) Despite the precise search query, 11 references were discarded due to reasons explained as follows. In chronological order, the first reference discarded was Papathoma-Kohle et al., (2019) because they use variables in the physical dimension, rather than socio-economic one. Two references from Yuan et al., (2019a, b) were identified by the search query as using the same method for the spatial assessment of SEV; so, we decided to select only one of them. Zhang and Huang (2018) address the topic of SV but not its spatial assessment, while Shen et al. (2018) focused on calculating the impact of disasters, rather than estimating SEV. The paper written by Goncalves, M., & Vizintim, M. F. B. (2017) was written in Portuguese, which none of the authors is proficient. Postiglione et al., (2016) promote a culture of seismic risk prevention, rather than to estimate SEV due to earthquakes. Alcántara-Ayala and Oliver-Smith (2014) present the activities undertaken by the ICL Latin -American network (ICL LAB) related to capacity building to reduce risk due to landslides, with no specific emphasis on SEV. Khazai et al., (2014), in their book chapter, concentrate on modelling shelter needs and health impacts caused by earthquakes. Vilches et al. (2014) evaluate the socio-environmental effects of the 27/10/2010 tsunami in Chile, considering the SEV among other aspects, but they do not make use of any spatial variable, indicator, or index, which is similar to the vulnerability assessment relating to a tsunami in the Town of Tirua (Chile) undertaken by Jaque Castillo et al.,(2013) (…)'.

- ▪ The result of your paper analysis should be also improved, and your conclusions need to be underpinned more closely with your results.

Thank you very much for this observation. The result section was improved and our conclusions this time are totally supported by our findings. Please see the text of both sections below:

[revised manuscript text omitted]

**Reviewer 1**

Thank you very much for your observations. You kindly spent time delving into our manuscript, and we are grateful. We have used a colour code to answer your questions. Please find your comments in grey, and the respective answers in black. The corresponding paragraph in the paper is in dark blue.

**Comments**

▪ Using one of the leading databases on scientific journal publications, the authors performed an assessment of articles published between 2008 and 2018 addressing the social and economic dimensions of vulnerability. From originally 235 articles, 21 were finally chosen for a detailed analysis. However, while the authors initially stated to undertake a systematic review for an application on an urban scale, the results and conclusions do not necessarily mirror this aim.

Thank you for your observation. A systematic review searches for, appraises and synthesises research evidence (Grant and Booth, 2009). Initially, the time period selected for the systematic review was 2008-2018, but as we are interested in elucidating the state of the art in data sources, spatial variables, indicators, methods, indexes and tools for the spatial assessment of the socio-economic vulnerability (SEV) related to geohazards in urban environments we decided to re-run the query updating also the period from 2010 to 2020. This time we undertook the search not only in Clarivate but also in two more databases: Scopus/Elsevier and Google scholar. Following the request of the reviewers we decided to include in this revised version the complete list of inclusion and exclusion search terms, that was not included in the previous version as a suggestion of one of the co-authors. Please find the search terms considered in Table 1.

| D | Q | SEARCH TERMS |
|---|---|---|
| Clarivate analytics | TOPIC | "social vulnerability" OR "economic vulnerability" OR "socioeconomic vulnerability" OR "socio-economic vulnerability" |
| | | AND |
| | TOPIC | "area" OR "distance" OR "range" OR "distance" OR "direction" OR "spatial geometries" OR "patterns" OR "spatial connectivity" OR "isolation" OR "diffusion" OR "spatial association" OR "scale" OR "accessibility" OR "network" OR "cluster" |
| | | AND |
| | TOPIC | "earthquakes" OR "tsunamis" OR "volcanic eruptions" OR "landslides" OR "soil erosion" OR "land degradation" |
| | | NOT |
| | TOPIC | "climate change" OR "ecological" OR "drought" OR "resilience" OR "debris" OR "epidemiological" OR "substance" OR "behavioural" OR "evacuation" OR "recovery" OR "pollution" OR "leptospirosis" OR "violence" OR "illness" OR "disease" OR "heat" OR "crisis" OR "conflict" OR "deaths" OR "obesity" OR "criminal" OR "chemical" OR "symptoms" OR "syndrome" OR "food insecurity" OR "air pollution" OR "stress" OR "diabetes" OR "depressive" OR "alcohol" OR "cancer" OR "drugs" OR "palm oil" OR "tobacco" OR "smoke" OR "storm" OR "psychometric" OR "cocaine" OR "toxic" OR "palliative" OR "therapy" OR "HIV" OR "dengue" OR "ecosystem" OR "rheumatoid" "arthritis" OR "nutritional" OR "malaria" OR "resources" OR "sexual activity" OR "sexual health" |
| | Art | |

| D | Q | SEARCH TERMS |
|---|---|---|
| Scopus/Elsevier | | (TITLE-ABS-KEY ( "social vulnerability*" AND "economic vulnerability*") AND TITLE-ABS-KEY ("socioeconomic vulnerability*") AND TITLE-ABS-KEY ("area" OR "distance" OR "range" OR "distance" OR "direction" OR "spatial geometries" OR "patterns" OR "spatial connectivity" OR "isolation" OR "diffusion" OR "spatial association" OR "scale" OR "accessibility" OR "network" OR "cluster") AND TITLE-ABS-KEY ("earthquakes" OR "tsunamis" OR "volcanic eruptions" OR "landslides" OR "soil erosion" OR "land degradation") AND NOT TITLE-ABS-KEY ("climate change" OR "ecological" OR "drought" OR "resilience" OR "debris" OR "epidemiological" OR "substance" OR "behavioral" OR "evacuation" OR "recovery" OR "pollution" OR "leptospirosis" OR "violence" OR "illness" OR "disease")) AND DOCTYPE (ar) AND PUBYEAR > 2009 AND PUBYEAR <2021 |

D: Database
Q: Query
Table 1. Terms included and excluded to select relevant literature references in Clarivate analytics.

Google scholar was discarded as a source of references given that when the query was run on this database, it gave us only one reference, which was the first version of this manuscript. Please see the evidence below in Figure 1.

[Figure]

Figure 1. Evidence of the result of the search query on the Google Scholar database.

- From their final choice of contributions, the main conclusion of the authors is that for assessing social vulnerability it is not sufficient to only compute a specific level of vulnerability, but also to include other spatial information available in order to avoid the modifiable areal unit problem (e.g., Unwin 1996).

Thanks for suggesting us this reference:

Unwin, D. J. (1996). GIS, spatial analysis and spatial statistics. Progress in Human Geography, 20(4), 540-551. doi:10.1177/030913259602000408

The section of conclusions has been revisited. Then, we invite you to check the new version. Please read below:

**5 Conclusions**

Based on the evidence, we can state that most of the spatial assessments of SEV in urban environments have been done for earthquakes and landslides and that Indonesia, China, Iran, and the USA lead the research in spatial assessment of SEV related to geohazards in urban environments. The scale of the spatial level of assessment – namely global, continental, subcontinental, national, regional, provincial, municipal, or local – determines the type of data to be collected and the assessment approaches. Although there have been advances, census data continues to be the most frequent source of data for the SEV assessments; however, in the case of spatial assessment, satellite images are now the main data source, facilitating the inclusion of the spatial component in SEV assessments. The spatial assessment of SEV allows visualising and communicating social phenomena and components that influence the degree of vulnerability that are not visible with other methods. The lack of data availability hinders the understanding of the concept of vulnerability (Zhou et al., 2014) and that is why VGI is essential today to obtain updated information in real-time at local scale, when other data sources are not available.

Traditional spatial variables and indicators continue to be used by authors, but combined with new variables, categories, and indicators, including the temporal dimension (day-night), and assessing at the local level, can increase the accuracy of spatial assessments of SEV and reduce uncertainty on their assessment. Each method for the spatial assessment of SV is selected according to the research aim, case study area, scale to cover, reliability of data sources, spatial variables and indicators available; geohazard to address, the scope of the research, and the level of funding. Methods such as ANN are gaining ground in the assessment of SEV. Other methods such as dasymetric population mapping enable more accurate SEV assessment. Factor analysis continues to be a useful tool to define the level of SEV based on primary dimensions and variables. Multi-criteria evaluation method offers a robust decision-making technique based on flexible choice and combination in criteria (Alcorn et al., 2013). SMCE incorporates the spatial component to the MCE to integrate spatial and non-spatial data to generate maps with multiple scenarios (Hizbaron et al., 2012). Classic methods such as FA are combined with more innovative ones such as ANP and fuzzy numbers to generate hybrid methods such as F'ANP. These new methods encourage the development of more complex hybrid methods such as F-ADP that increase the accuracy and reduce the uncertainty levels in the spatial SEV assessments. Ordinal logistic regression and binary logistic regression are useful methods to identify spatial variables as determinants of SEV. The spatial component can be also be added by simply overlapping the areas with high SEV with hazard zones using GIS. Most authors have built upon the SoVI® developed by Cutter et al. (2003) to quantify SEV or to create their own SEV indexes, demonstrating that it remains the benchmark for the assessment of SEV and a reference for its spatial assessment.

Geographic Information Systems, statistical analysis, RS, programming languages, and interactive databases are the tools currently used by the scientists for the assessment of SEV vulnerability. The spatial assessment of SEV in the areas where it is requested must depend not only on the financial resources for research but also on the availability of opensource software with the functionalities of spatial statistics, such as QGIS, GeoDa or Ilwis. Authors combine traditional and new data sources, spatial variables and indicators, methods, indexes and tools including the temporal dimension, increasing the resolution to the local level with the aim to increase the accuracy and reduce the uncertainty of spatial assessments of SEV related to geohazard in urban environments.

- The overall choice of keywords and exclusion of other keywords results in the fact that many studies addressing social vulnerability and/or economic dimensions of vulnerability have not been considered by the authors, which in turn restricts the overall conclusions possible.

Thanks for your comment. We agreed that there are many studies addressing SEV and/or economic dimensions, but not all of them consider the spatial dimension, which is the main aim of this research. The conclusions have been revisited as you could check in the previous section. The 23 references finally reviewed are listed in Table 2.

[revised manuscript text omitted]

- The time period covered is not justified

Thank you for this observation. The reason for selecting the period 2008–2018 and update the period to 2010-2020 for this revised version was to explore the state of the art on data sources, spatial variables, indicators, methods, indexes and tools for the assessment of SEV related to geohazards in urban environments which we believe can be covered in the past 10 years. We included this clarification in the methods section of the revised version of the manuscript. Additionally, in the introduction and the discussion section, we included relevant references before this period following a request of the editor. Please find below the correspondent text below:

'(…) A systematic review searches for, appraises, and synthesises research evidence (Grant and Booth, 2009). In the present research, the systematic review was conducted to elucidate the state of the art of data sources, spatial variables, indicators, methods, indexes and tools for the spatial assessment of the SEV related to geohazards, which we consider is covered in the period between 2010 and 2020. Thus, the main research question is: what is the state of the art in the spatial assessment of SEV to geohazards in urban environments? (…)'

- The overall aim to provide a structured overview on studies and indicators, which is not only promised in the title of the contribution but also in the introduction, is not mirrored by the main text body. Materials, methods and findings are rather compiled in a very unstructured way which makes a structured conclusion quite challenging.

Thank you for your observation. We are afraid that we do not have a materials section. The data sources, spatial variables, indicators, methods, indexes and tools, which we believe you named 'findings', are listed in tables. The structure of the tables is explained in the methods section. Please find the explanation below:

[revised manuscript text omitted]
 (page 2, lines 15/16), which according to my experience is exactly the contrary relationship – also here we do have scholarly articles which did not make it to the current overview. One reason is again the choice of keywords (see below).

Thanks for the comment. However, while we respect your opinion, based on the previous work of one of the authors, we prefer to stick with the definition of economic dimension of vulnerability formulated by Birkmann et al. (2013 p. 200): 'Economic dimension: propensity for loss of economic value from damage to physical assets and/or disruption of productive capacity'. We consider the opposite concept is more related to economic resilience than vulnerability.

▪ In the introduction it becomes not clear which specific research question should be answered and were the niche and the gap for the contribution is to be found.
Thank you for your observation and question. The research question is formulated in the methods section and it is: '(…) what is the state of the art in the spatial assessment of SEV to geohazards in urban environments? (…)'

▪ Paragraphs addressing common sense are somehow not connected to those showing specific issues; to give an example it remains unclear why paragraph 3 immediately starts with the SoVI as one of the indices available to assess social vulnerability.

Thanks for this observation. The structure of the paper was modified with the aim to solve this kind of problems. We expect that the comment above does not apply anymore.

On page 3, line 18 the authors even conclude (or state) that only a few authors have elaborated on the spatial dimension of social vulnerability, which is wrong if proper literature research would have been undertaken. There are lots of studies around on this topic, some of them in the target journal NHESS.

Thanks for this comment. Based on the new search query, we found that the reviewer was totally right, in consequence, we decided to delete this sentence.

Moreover, the statement that spatial vulnerability assessments only became prominent after the 2004 Indian Ocean Tsunami is neither proven by references, nor true according to my own knowledge.

Thank you for this observation about an incorrectly drafted sentence. Citing Fekete (2012), we wanted to state that events such as the Indian Ocean tsunami in 2004 and Hurricane Katrina in 2005, each with an explicit spatial component, sparked again the research community's interest in those social groups that are more affected by this type of phenomena. We rephrased the sentence accordingly:

'(…) The Indian Ocean tsunami in 2004, as a result of its large impact area, reignited the research community's interest in spatial vulnerability analyses, illuminating the problems faced by low-income population after disasters (Fekete, 2012). This approach was aligned with the Hyogo Framework for Action (UNISDR, 2007), and confirmed by Gautam (2017), who notes that after 2005 a focus on construction and mapping of the SV index intensified. Thus, the use of geographic information systems (GIS) to collect and process data related to hazards and vulnerability was found very suitable (Fekete, 2012). Major earthquakes in the same period as this systematic review (2010-2020), e.g. Chile (2010), New Zealand (2010 and 2011), Nepal (2015), Mexico (2017), Albania (2019), and Croatia (2020) demonstrate the vulnerability of urban areas to seismic damages (Armaş et al., 2017) (…)'.

Further, the use of GIS is not only very suitable for assessing spatially the issues of social and economic vulnerability, but it is also a tool to exactly do this.

Thank you for this observation about use of GIS. While we agree with your claim, after carefully going through the manuscript, we found that the exact statement in the manuscript is (line 4, page 4): '(…) Thus, the use of geographic information systems (GIS) to collect and process data related to hazards and vulnerability was found very suitable (Fekete, 2012) (…)'. The reason for including this statement is that, in the past, hazard and vulnerability data collection processes were performed manually, making the assessment highly time-consuming. Now, these hazard and vulnerability assessments are speeded up with the integration of GIS into the process. However, the potential of GIS is sometimes untapped and limited to the mapping of the socio-economic characteristics of a case study area, without taking into account the influence of the spatial component that can be integrated to take advantage of the GIS capabilities.

Finally, the statement that vulnerability is dynamic and subject to spatial and temporal dynamics across scales is not very innovative, there are even specific research papers on this topic from the period 2008-2018.

Thank you for this comment. We agree with the reviewer, in consequence, the statement was deleted.

- Methods: it remains totally open how the amount of 235 papers initially identified was reduced to the final set of 21 contributions.

Thanks for your observation. You can follow the explanation on the text as follows:

'(…)The gross number of articles identified using the search query were 29, having two matching references in Clarivate Analytics and Scopus/Elsevier: Kurnianto et al., (2019) and Eidsvig (2014). Thus, eventually, we identified 27 references. Despite the precise search query, 11 references were discarded due to reasons explained as follows. In chronological order, the first reference discarded was Papathoma-Kohle et al., (2019) because they use variables in the physical dimension, rather than socio-economic one. Two references from Yuan et al., (2019a, b) were identified by the search query as using the same method for the spatial assessment of SEV; so, we decided to select only one of them. Zhang and Huang (2018) address the topic of SV but not its spatial assessment, while Shen et al. (2018) focused on calculating the impact of disasters, rather than estimating SEV. The paper written by Goncalves, M., & Vizintim, M. F. B. (2017) was written in Portuguese, which none of the authors is proficient. Postiglione et al., (2016) promote a culture of seismic risk prevention, rather than to estimate SEV due to earthquakes. Alcántara-Ayala and Oliver-Smith (2014) present the activities undertaken by the ICL Latin -American network (ICL LAB) related to capacity building to reduce risk due to landslides, with no specific emphasis on SEV. Khazai et al., (2014), in their book chapter, concentrate on modelling shelter needs and health impacts caused by earthquakes. Vilches et al. (2014) evaluate the socio-environmental effects of the 27/10/2010 tsunami in Chile, considering the SEV among other aspects, but they do not make use of any spatial variable, indicator, or index, which is similar to the vulnerability assessment relating to a tsunami in the Town of Tirua (Chile) undertaken by Jaque Castillo et al.,(2013). Five references from the previous search query carried out in 2018, and not identified in the refined search query, were included in the list given their relevance due to the geohazards and spatial variables, indicators, and indexes that they address. (…)'.The scheme of the methodology applied is depicted in Figure 1.

[Figure]

Figure 1. Methodology applied for the systematic literature review

- Moreover, searching only for combinations of "social vulnerability" excludes the amount of (valuable) papers around addressing multiple dimensions of vulnerability – and some of these contributions again can be found in NHESS.

Thank you for comment. We are aware that apart from the social and economic dimensions, other dimensions of vulnerability also exist such as physical, cultural, environmental and institutional (Birkmann et al., 2013). Nevertheless, we prefer to focus on the social and economic dimensions of the vulnerability for this specific research.

- Further, the authors state in the text that they excluded terms such as "climate change", "health" and "crime analysis", whereas in Figure 1, much more terms have been excluded. BTW: Why has the term "debris" been excluded? Just to give an example, many studies on (social and economic) vulnerability are related to dynamic flooding such as flash floods and debris flows/torrential hazards (even the mentioned EU-funded project MOVE), these are completely ignored by the authors due to their choice of key words.

Thank you for your comment. We are afraid that the term 'debris' suggested by you is not a spatial variable, indicator or index useful for the assessment of SEV.

- In contrast, some of these hazard types are then mentioned in the results section (page 7, second paragraph).

Thanks for your observation. In the revised version of the manuscript, we decided to focus sharply on the topic of the spatial assessment of SEV related to geohazards, therefore this comment does not apply anymore.

- Instead of showing which contributions used which methods or indicator groups for assessment, the authors could have shown the challenge of indicator interdependencies, one of the main points of criticism for the SoVI. Simply applying the SoVI does not necessarily result in an overview on social and economic dimensions of vulnerability because of the inherent dependencies between indicators.

Thank you very much for this suggestion. Nevertheless, the research question of the manuscript is: what is the state of the art in the spatial assessment of SEV to geohazards in urban environments?, rather than discuss the challenge of indicator interdependencies. However, when we discuss the current methods used for the selection of spatial indicators, we are indirectly addressing this topic. Please read the text below as it is written in the manuscript:

'(…) This systematic review identified the versatility of ANN, which can be either used to extract monthly rainfall data (Aksha et al., 2020), for deriving social vulnerability maps (SVM) (Alizadeh et al., 2018) or to train the self-organized map (SOM) algorithm cluster method (Maharani et al., 2016). The use of dasymetric population mapping not limited to administrative boundaries, even going to block-level to increase the spatial resolution of the population exposure analysis (Garcia et al., 2016) and additionally by including the temporal dimension with its day-night variability, enables improving the accuracy of the spatial assessments of SEV (Yuan et al., 2019a). Factor analysis (FA) is used by Castro et al. (2015) to establish the level of SEV and by Zebardast (2013) to extract primary dimensions and variables of SEV. Alcorn (2013) applied MCE to assess economic vulnerability using four significant factors: population, infrastructure, land use, and economic production. SMCE is applied by Armaş et al., (2017) to integrate social, education, housing, and social dependence vulnerability dimensions and by Hizbaron (2012) to develop

deterministic SV scenarios. Zebardast (2013) enters the variables of SEV into a network model in an analytic network process (ANP) to rank the importance of each variable to complete the F'ANP method. This method is focused on developing a composite social vulnerability index (SOVI). Binary logistic regression was the statistical method applied by Qasim et al. (2018) to identify the determinants of landslide risk perception, location being one of them. Walker et al., (2014) present a multi-criteria evaluation (MCE) model that incorporates access to healthcare facilities using GIS to identify and rank residential areas in Victoria, British Columbia. The integration of the concept of uncertainty into ANP using fuzzy numbers (F-ANP) is combined by Rezaei-Malek et al., (2019) with fuzzy DEMATEL (F-DEMATEL) to deal with the interdependency among a set of criteria and fuzzy PROMETHEE II (F- PROMETHEE II) to control the criteria weights, the complete method is denominated fuzzy ANP DEMATEL PROMETHEE II (F-ADP). Ordinal logistic regression (OLR) is used by Muir et al., (2019) to predict the mental health condition of people displaced by series of volcanic eruptions in Merapi, Indonesia, according to their migration status (displaced, moved home, in transition, and moved on), which implies a spatial component. Geological experience and logical analysis method were used by Chen (2016) to select indicators. Toke et al.,(2014) undertake an overlay analysis to identify the census block groups that intersect zones with an extreme ground shaking hazard (…)'.

- In the discussion section the authors have raised some issues that remain questionable, such as the fact that most of the articles related to flood hazard and social vulnerability have been written by geographers because they may be interested in environmental vulnerability.

Thanks for this observation. This statement is not anymore in the revised version of the manuscript since it tackles the spatial assessment of SEV related only to geohazards.

- These issues are just examples underpinning the overall judgement that this contribution is so far not up to international standards. Although the authors have some interesting arguments, I believe that the manuscript needs further improvement to bring it up to an acceptable level before it can be accepted for publication.

Thanks for the assessment of our manuscript. We expect that based on your comments, the comments of the second reviewer and the editor, we have produced a revised version that meets international standards and can, therefore, be published.

- To summarise, it is not clear why the authors chose specific keywords and excluded others, it is not clear why the authors chose the distinct time period between 2008 and 2018 (the discussion on multiple dimensions of vulnerability and the spatiality of vulnerability is much older). The results are not presented in a logical and organised manner, and the conclusions are not underpinned by the results, some of them seem rather driven by speculation than by evidence.

Thank you to you and the reviewers for the detailed assessment of our manuscript. We selected terms and exclude others to answer our research question: what is the state of the art in the spatial assessment of SEV to geohazards in urban environments? which we consider is covered in the period between 2010 and 2020. The aim of this manuscript was not to elaborate on the multiple dimensions of vulnerability, but rather to focus on the social and economic dimensions. We agree that the discussion of the spatiality of vulnerability started long ago and that is why some of the spatial variables were extracted from Meentemeyer (1989). We have restructured our results, and the conclusion section was totally rewritten based on evidence to avoid speculations.

Again, we are most appreciative of the attention and care you have given to our manuscript, and we hope you will now consider it for minor revisions and eventually for publication.

[revised manuscript text omitted]

**Moved up [5]:** The most imp... data. This factor can lead to reli... most accurate indicators of vuln... Furthermore, considering the m... assessment of socio-economic v... indicators are selected and index... context, we consider the SV ind... and the SVU index developed b... benchmarks of what a spatial ind... vulnerability in the urban conte... could include the Normalised D... and spatial indicators, such as ro... customer proximity, included in... (2013). The Walk Score®, deve... although originally orientated to... walkability, could be adapted as... validate any spatial index for as...

**Moved down [4]:** The prese...

**Moved down [9]:** ¶

**Commented [B1]:** Which c...

[revised manuscript text omitted]

---

## Author Response (AR2)

Dear Editor,

Thank you very much for the acknowledgement to our effort and the result reflected in the manuscript. We are going to go carefully through the aspects indicated by the reviewers to continue improving the manuscript, with a special focus on the figures.

Best regards,

Diana Contreras

**Reviewer 1**

Thank you very much for your observations. You kindly spent time again delving into our manuscript, and we are grateful. We have used a colour code to answer your questions. Please find your comments already solved in grey, and the respective answers in black. The corresponding paragraphs in the paper are in dark blue.

**Comments**

Using the two leading databases on scientific journal publications (Scopus/Elsevier and SCI/Clarivate Analytics), the authors performed an assessment of articles published between 2010 and 2020 addressing the social and economic dimensions of vulnerability. A number of 27 publications were finally chosen for a detailed analysis on data sources, spatial variables, indicators, methods, indexes and tools for the spatial assessment of socio-economic vulnerability (SEV) related to geohazards. In the revised version, the paper reads much clearer and is more concise so that the potential readers can follow the string of argumentation as well as (by the included Tables) the relevant indicators discussed.

Overall, the manuscript is therefore suitable for publication in the target journal. My comments on the first version of the manuscript have been thoroughly acknowledged. The authors may wish to additionally specify the following:

- Page 1, line 28: my recommendation here is to refer to the original UNDRR source and not to Aubrecht et al. (2013) – this statement can be found in nearly all publications related to the topic, and I strongly recommend to cite the original source here.
  Thank you very much for your recommendation. Please find the change accordingly:

  '(…) Vulnerability is defined by United Nations (UN) as 'the conditions determined by physical, social, economic and environmental factors or processes which increase the susceptibility of an individual, a community, assets or systems to the impact of hazards' (UN, 2016) (...)'.

- Page 5, line 15: I would recommend to insert "Austria" after Salzburg for those readers not familiar with European cities.
  Thank you very much for your recommendation. Please find the change accordingly:

  '(…) In the case study area of Salzburg (Austria), an expert-based approach was chosen, and several experts were asked to allocate weights according to the contribution of each variable to the vulnerability of floods (Contreras and Kienberger, 2011) (…)'.

- Page 6, lines 8ff: I would prefer to use "endogenic" and "exogenic" geohazards instead of "internal" and "external", to be in line with the geoscience literature (also page 7, line 21).
  Thank you very much for enlightening us with regards to the appropriate terms to use; we appreciate this insight. Please find the changes accordingly:

'(…) Geohazards can be endogenic such as earthquakes, tsunamis, and volcanic eruptions and exogenic such as landslides, soil erosion, and land degradation (…)'.

'(…) The terms selected for the search query refer to vulnerability in the socio-economic dimension, the spatial variables listed by Meentemeyer (1989), Béné (2009), Contreras et al. (2013) and Buzai and Villerías Alarcón (2018) and the aforementioned endogenic and exogenic geohazards (…)'.

'(…) The lack of articles that tackle exogenic geohazards can be explained by the fact that we excluded from the search query words such as "climate change" OR "ecological" OR "drought", which are indirectly related to these phenomena (…)'.

- Moreover, the argument that "…we chose these hazards because they produced high losses… in Chile" seems a bit strange in the context of this manuscript, and can simply be deleted. If it is the choice of the authors to focus on the hazard types indicated, from my point of view, it is not necessary to state the reason. For the overall review it is not important whether or not certain hazard types chosen are relevant in Chile, it simply detracts the reading flow.

  Thank you very much for your comment. As we stated, the reason to choose geohazards phenomena came as a request from a previous reviewer. Therefore, we agree to delete the reference to Chile, but we must keep the remainder of the text that explains the reason to choose geohazards phenomena.

  '(…) We particularly focus on these phenomena for two reasons: first, geohazards are the natural phenomena that have produced the highest quantity of losses in recent years in the urban environments, particularly earthquakes, and, second, because geohazards are the phenomena addressed by the institutions involved in the present research (…)'.

- Besides, I still have a certain doubt about leaving out other geohazards of exogenic type which – in particular in the US literature – are often subsumed under the term "landslides", such as the "debris flow"-type hazard group, or the entire group of (gravitational) mass movements – please see e.g. the recent publication of Mazzorana et al. (2019, page 1092) explicitly focusing on Chile. Nevertheless, given the statements made in these lines it is ok for me just to name the hazard groups that were analysed, which in turn explains why other hazard types also relevant (for Chile) are missing.

  Thank you very much for your comment. We also expected to have more literature references related to landslides in our systematic review, but after running the query using the terms defined in Table 1 (please see below), we managed only to obtain 5 references, which is 17% of the references identified for this systematic review. We deleted the reference to Chile following your advice.

| Topic | (social vulnerability* OR societal vulnerability* OR socioeconomic vulnerability* OR socio-economic vulnerability* OR economic vulnerability*) |
|---|---|
| | AND |
| Topic | (area* OR distance* OR range* OR distance* OR direction* OR spatial geometries* OR patterns* OR spatial connectivity* OR isolation* OR diffusion* OR spatial association* OR scale* OR accessibility* OR network* OR cluster*) |
| | AND |
| Topic | (earthquakes* OR tsunamis* OR volcanic eruptions* OR landslides* OR soil erosion* OR land degradation) |
| | NOT |
| Topic | (climate change* OR ecological* OR drought* OR resilience* OR debris* OR epidemiological* OR substance* OR behavioral* OR evacuation* OR recovery* OR pollution* OR leptospirosis* OR violence* OR illness* OR disease* OR heat* OR |

> crisis* OR Conflict* OR deaths* OR obesity* OR criminal* OR chemical* OR symptoms* OR syndrome* OR food insecurity* OR air pollution* OR stress* OR diabetes* OR depressive* OR alcohol* OR cancer* OR drugs* OR palm oil* OR tobacco* OR smoke* OR storm* OR psychometric* OR cocaine* OR toxic* OR palliative* OR therapy* OR HIV* OR dengue* OR ecosystem* OR rheumatoid arthritis* OR nutritional* OR malaria* OR resources* OR sexual activity* OR sexual health*).

Table 1. Terms included and excluded to select relevant literature references

Literature references related to landslides identified in our systematic review using the terms listed in Table 1. Please see below:

- Chen, Y. (2016). Conceptual Framework for the Development of an Indicator System for the Assessment of Regional Land Subsidence Disaster Vulnerability. Sustainability, 8(8), 757.
- Ley-García, J., Denegri de Dios, F. M., & Ortega Villa, L. M. (2015). Spatial dimension of urban hazardscape perception: The case of Mexicali, Mexico. International Journal of Disaster Risk Reduction, 14, 487-495. doi:https://doi.org/10.1016/j.ijdrr.2015.09.012
- Toke, N. A., Boone, C. G., & Arrowsmith, J. R. (2014). Fault zone regulation, seismic hazard, and social vulnerability in Los Angeles, California: Hazard or urban amenity? Earths Future, 2(9), 440-457. doi:10.1002/2014ef000241
- Zeng, J., Zhu, Z. Y., Zhang, J. L., Ouyang, T. P., Qiu, S. F., Zou, Y., & Zeng, T. (2012). Social vulnerability assessment of natural hazards on county-scale using high spatial resolution satellite imagery: a case study in the Luogang district of Guangzhou, South China. Environmental Earth Sciences, 65(1), 173-182. doi:10.1007/s12665-011-1079-8
- Ebert, A., Kerle, N., & Stein, A. (2009). Urban social vulnerability assessment with physical proxies and spatial metrics derived from air- and spaceborne imagery and GIS data. Natural Hazards, 48(2), 275-294. doi:10.1007/s11069-008-9264-0

Thanks for the recommendation of the reference:
- Mazzorana, B., Picco, L., Rainato, R., Iroumé, A., Ruiz-Villanueva, V., Rojas, C., Valdebenito, G., Iribarren-Anacona, P., and Melnick, D.: Cascading processes in a changing environment: Disturbances on fluvial ecosystems in Chile and implications for hazard and risk management, Science of the Total Environment, 655, 1089-1103, 2019.

However, given that this reference does not address any of the geohazards included in this systematic review, we feel that we should not include it in this paper. However, we do appreciate the suggestion and will keep it in mind for future research studies.

**Reviewer 2**

Thank you very much for the acknowledgement to our effort to improve the manuscript, we highly appreciate your insights. You kindly spent time again delving into our manuscript, and we are grateful. Please find your comments already solved in grey and the respective answers in black. The corresponding paragraph in the paper is in dark blue.

**Comments**

Very good and detailed improvements, impressive energy, time and very detailed documentation of changes made. Congratulations for such efforts.

▪ Make another technical check - some typos.

Thank you very much for this recommendation, we went again carefully through the manuscript and still, we asked for additional support of our English proof-reader to be sure that we eliminate all possible typos, grammar or punctuation mistakes on the manuscript.

o 'society and the economy are only two of the dimensions of vulnerability (…)'.

o '(…) Data constraints play a key role in the results of the SEV assessment (…)'.

o '(…) The construction of an index implies selection of indicators, indicator normalization and weighting, and aggregation into an index (OECD, 2008) that must collectively represent aspects of a society's ability to prepare for, deal with and recover from a disaster (…)'.

o '(…) We limited the query to articles published in academic journals because they typically are rigorous in the selection of their publications and therefore contain a complete and accurate description of methodologies and consistent results (…)'.

o '(…) To prioritize disaster-prone areas that are known as potential demand points (PDPs) given their vulnerability under large-scale earthquakes (…)' .

o '(…) To apply an artificial neural network (ANN) and geographic information system (GIS) for estimating the social vulnerability to earthquakes in the Tabriz city, Iran (…)'

o 'For the purpose of the systematic review, we found that the Clarivate Analytics database more accurately identified the references for this systematic review, and it is more user-friendly than other databases (…)'.

o '(…) Housing quality and tenancy conditions describe the vulnerability of the population to become homeless after a disaster (Toke et al., 2014) (…)'.

o '(…) Accessibility as a spatial indicator is defined as the ability to contact and interact with places of economic or social opportunities (Deichmann, 1997). (…)'

o '(…) Alcorn et al. (2013) used an improved version of the same index but specifically adapted it to the variability in SEV in the case study area that was focused on census-designated places (CDPs) on a small scale. (…)'

o '(…) Toké et al (2014), build upon the SoVI® to create their own SV indexes that incorporate the spatial dimension. According to the LA-SoVIC developed by Toket et al. (2014), SV is highly linked to the normalised difference vegetation index (NDVI) as a proxy for urban green space (…)'.

o '(…) Armaş et al., (2017) applied a pairwise comparative method in the AHP implemented in the SMCE module of the Integrated Land and Water Information System (IlWIS) software (…)'.

o 'Based on the evidence, we can state that most of the spatial assessments of SEV in urban environments have been done for earthquakes and landslides and that Indonesia, China, Iran, and the USA lead the research in the spatial assessment of SEV related to geohazards in urban environments (…)'.

o 'The lack of data availability hinders the understanding of the concept of vulnerability (Zhou et al., 2014) and that is why VGI is essential today to obtain updated information in real-time at the local scale when other data sources are not available (…)'.

o '(…) The spatial assessment of SEV in the areas where it is requested must depend not only on the financial resources for research but also on the availability of opensource software with the functionalities of spatial statistics, such as QGIS, GeoDa or IlWIS (…)'.

o '(…) Likewise, spatial assessments of SEV must be considered before taking resettlement decisions for not creating again spatial conditions that favour the SEV (…)'.

▪ Make sure figures are provided in high resolution at the end.
Thank you very much for your observation. The high-resolution images were provided to the editor in separate files in the previous version. This time the high-resolution images (TIFF/300dpi) will be included in the text.

▪ Figure 1. may be hard to read due to colour red and rather thin font.

Thank you very much for your observation. This time, we have included a high-resolution image (TIFF/300dpi) in the text; we also increased and changed the font from Arial to Times New Roman to improve the readability of the figure for the reader. All the text is in bold. The colour, rather than red, is orange (68R/68G/72B). We selected this colour for two reasons, on the one hand, it is an eye-catching colour; on the other hand, it is a colour in the corporate image of the Research Centre for Integrated Disaster Risk Management (CIGIDEN), one of the institutions involved in this research. In case the editor considers that it is not a proper colour, we are willing to change it to grey (255R/55G/0B). Please see the result below:

[Figure]

Figure 1. Methodology applied for the systematic literature review.

- Figure 2: Font size may be too small in final published version to read easily for terms under columns.

Thank you very much for your observation. The font size of the terms under the bars (geohazards) under the column was increased from 9pt to 11pt.

[Figure]

Figure 2. Number of literature references in the systematic review that addresses each geohazard

- Table 3: Check if this is intended that second column sometimes directly starts with author names. To me this is rather confusing.

Thanks for your question. The second column in Table 3 lists, in some cases, the specific name of the census, the kind of satellite images, the name of the disaster databases, the name of the land use/land cover (LULC) maps, and the name of population datasets used by the authors. This detail is not necessary in the case of the other data sources; that is why, in these cases, the second column starts directly with the author names. We explain this aspect in the manuscript as follows:

'(…) The second column in Table 3 includes, in some cases specific details of the data source used by the authors (…)'.

- Figures 2&3 are nice, but I am a bit doubtful whether they actually do not raise more questions such as how much is overpresence of earthquakes due to selection method / search terms or due to occurrence. And the title of your paper is about urban contexts - so are these all truely (spatial) urban studies? If so, would maybe a list of cities reveal more details than the world map - like are certain countries always having 1-2 cities presented mainly, while in other countries, it is 3-4 cities. That could another interesting insight. The map as such rather prompts to think about the representativeness of just about 20 analysed papers. As does Figure 2. But I am also fine if you keep both figure 2 and 3 as they are.

Thanks for this comment. The literature references identified are based on a highly detailed search query to avoid bias. The query could be repeated any time, and the results will be always the same although additional publications from 2020 could appear in the result. Rather than go deeply in highlighting specific countries leading, and case study areas used in the research on spatial assessment of socio-economic vulnerability (SEV), we only wanted to include an additional finding supported by the research undertaken by Shen et al., (2018). We prefer to focus on the main research question of this manuscript: What is the state of the art in the spatial assessment of SEV to geohazards in urban environments? The state of the art is not necessarily linked to specific case study areas.

The total number of references reviewed were many more than 20. In the first version of the manuscript, based on a previous more general query not focused on geohazards, we identified 235 literature references, from which we found 84 relevant references, and 42 highly relevant references. Finally, 21 references were selected for review. From the previous search and given their relevance. six of these references were considered again in the current version of the manuscript. In the current version, we carefully reviewed all 29 references, but eventually we selected 18 and ruled 11 out for the reasons already explained in the manuscript. The query is extremely specific, which explains the reduced number of references identified, but the literature reviewed in this research included many more than the 24 selected at the end. Additionally, the result in the previous search regarding case study areas was similar, with the USA and China leading the research in the spatial assessment of SEV. As a result, we prefer to keep the Figures 2 and 3 as they are, and as you agree also with this decision, we made some changes in the format as you also requested. Please find below the new version of Figure 3. We added this explanation in the discussion section of the manuscript. Please read below:

[revised manuscript text omitted]